METHODS AND RESOURCES

# A generic cell surface ligand system for studying cell–cell recognition

**Eleanor M. Denham**[1], **Michael I. Barton**[1], **Susannah M. Black**[1], **Marcus J. Bridge**[1], **Ben de Wet**[1], **Rachel L. Paterson**[1], **P. Anton van der Merwe**[1]☉*, **Jesse Goyette**[1,2]☉¤*

**1** Sir William Dunn School of Pathology, University of Oxford, Oxford, Oxfordshire, United Kingdom, **2** EMBL Australia Node in Single Molecule Science, School of Medical Sciences, University of New South Wales, Sydney, New South Wales, Australia

☉ These authors contributed equally to this work.
¤ Current address: EMBL Australia Node in Single Molecule Science, School of Medical Sciences, University of New South Wales, Sydney, New South Wales, Australia
* j.goyette@unsw.edu.au (JG); anton.vandermerwe@path.ox.ac.uk (PAvdM)

**Data Availability Statement:** All relevant data are within the paper and its Supporting Information files.

## Abstract

Dose-response experiments are a mainstay of receptor biology studies and can reveal valuable insights into receptor function. Such studies of receptors that bind cell surface ligands are currently limited by the difficulty in manipulating the surface density of ligands at a cell–cell interface. Here, we describe a generic cell surface ligand system that allows precise manipulation of cell surface ligand densities over several orders of magnitude. These densities are robustly quantifiable, a major advance over previous studies. We validate the system for a range of immunoreceptors, including the T-cell receptor (TCR), and show that this generic ligand stimulates via the TCR at a similar surface density as its native ligand. We also extend our work to the activation of chimeric antigen receptors. This novel system allows the effect of varying the surface density, valency, dimensions, and affinity of the ligand to be investigated. It can be readily broadened to other receptor–cell surface ligand interactions and will facilitate investigation into the activation of, and signal integration between, cell surface receptors.

## Introduction

Many cellular receptors are activated by ligands presented on other cell surfaces. To study these receptors in depth, cell lines expressing the appropriate physiological ligand are required. Conducting dose-response experiments on these receptors is challenging as controlling ligand density on the surface of cells is difficult. Currently, this is limited to sorting cells into populations with varying expression levels, using inducible expression systems, or the use of blocking antibodies. Commonly used alternatives to physiological ligands are antibodies specific for a receptor or recombinant ligands presented on artificial surfaces, such as plastic culture dishes or supported lipid bilayers on glass coverslips. However, this can be a poor mimic for the complexity and biophysical characteristics of cell surfaces.

One widely studied group of receptors are non-catalytic tyrosine-phosphorylated receptors (NTRs), also called immunoreceptors, which are the largest group of receptors expressed on leukocytes [1]. These receptors play a major role in the recognition of infected or cancer cells.

**Funding:** This work was supported by two Wellcome Trust PhD Studentships (EMD, grant reference:097108/Z/11/Z; RLP, grant reference: 099812/Z/12/Z), a Wellcome Trust Senior Investigator Award (PAvdM, grant reference: 101799/Z/13/Z), a Nuffield Medical Fellowship from the Australian Academy of Science (JG, grant reference: #1016848), and a grant from the National Health and Medical Research Council of Australia (JG, grant reference: APP1163814). The funders had no role in study design, data collection and analysis, decision to publish, or preparation of the manuscript.

**Competing interests:** The authors have declared that no competing interests exist.

**Abbreviations:** CAR, chimeric antigen receptor; CHO, Chinese hamster ovary; CTLA-4, cytotoxic T-lymphocyte-associated antigen-4; DAP12, DNAX-activating protein of 12 kDa; eGFP, enhanced green fluorescent protein; FKBP, FK506 binding protein; FRB, FKBP-binding domain; HA, hemagglutinin; HEK293T, human embryonic kidney 293T; IL-8, interleukin 8; mCherry, monomeric Cherry; mTFP, monomeric teal fluorescent protein; NFκB, nuclear factor kappa-light-chain-enhancer of activated B cells; NK, natural killer; NTR, non-catalytic tyrosine-phosphorylated receptor; PD-1, programmed death-1; PDB, Protein Data Bank; pMHC, peptide presented in major histocompatibility complexes; s.f., significant figures; SCD, single-chain dimer; SEM, standard error of the mean; Siglec-14, Sialic acid–binding immunoglobulin-type lectin 14; SIRPβI, signal regulatory protein βI; TCR, T-cell receptor.

Since some NTRs, such as the T-cell receptor (TCR), cytotoxic T-lymphocyte-associated antigen-4 (CTLA-4), and programmed death-1 (PD-1), have been successfully manipulated/targeted for therapeutic purposes [2, 3], the remaining NTRs are currently under intense investigation for the development of immunotherapies (examples include [4–7]).

The mechanism of signal transduction, or triggering, has been studied in depth for some NTRs (notably the TCR) but remains controversial [1, 8]. Because of the widespread importance of NTR function in immune regulation, and huge interest in their activity, elucidating this mechanism is critical to both our understanding and ability to modulate receptor activity when required in clinical settings. Whereas NTRs have conserved signalling modules, their extracellular regions are rapidly evolving and hugely diverse and bind a structurally diverse range of ligands [1]. This diversity, and the fact that their ligands are often not known, has hampered a systematic investigation of NTRs.

These limitations motivated us to develop new tools for investigating receptors that have cell surface ligands. We present a novel generic ligand system whereby a single ligand can engage any receptor engineered to have an accessible tag. We show that ligand density can be varied easily and precisely quantified, and we show that the generic ligand can activate a number of representative NTRs in a clear dose-dependent manner. For one receptor, the TCR, we compare the response to generic versus physiological ligand: peptide presented in major histocompatibility complexes (pMHC).

We also describe and briefly show how this generic ligand can be manipulated to alter other biochemical and biophysical properties of receptor-ligand interactions such as valency, affinity, and dimensions. These modifications to a single ligand will apply to any interaction involving the ligand and a tagged receptor and thus permit high-throughput, systematic analyses of multiple receptors.

Finally, we show that this system can be adapted to study chimeric antigen receptors (CARs) and to present multiple ligands.

## Materials and methods

All data were fitted using GraphPad Prism or FlowJo software. In all cases in which sample values were below the machine detection limit, those samples were given a value of 0.

### Constructs

All sequences were verified by Sanger sequencing (Source BioScience or Eurofins Scientific).

**Strep-tag II/Twin-Strep-tag.** Sequences encoding the Igκ leader sequence, Strep-tag II or Twin-Strep-tag with linker (SAWSHPQFEK or SAWSHPQFEKGGGSGGGSGGGSAWSHPQFEK), a multiple cloning site, FLAG tag, and 3′ stop codon were inserted into the pHR-SIN-B-X-IRES-EmGFP lentiviral vector (a gift from Vincenzo Cerundolo, University of Oxford) using 5′ BamHI and 3′ NotI restriction sites and the GENEWIZ gene synthesis service. This insertion destroyed the plasmid BamHI site and removed the internal ribosome entry site–emerald green fluorescent protein (IRES-EmGFP) sequence: pHR-SIN-BX-Strep-tag-II or pHR-SIN-BX-Twin-Strep-tag, respectively.

**Receptors.** DNA encoding human signal regulatory protein βI (SIRPβI) (amino acids 30–398), Sialic acid–binding immunoglobulin-type lectin 14 (Siglec-14) (amino acids 17–396), or natural killer (NK) p30 (amino acids 19–201) with either 5′ BamHI or XhoI and 3′ BsiWI restriction sites were inserted into the multiple cloning site of pHR-SIN-BX-Twin-Strep-tag. DNA encoding Siglec-14 (amino acids 17–396) was likewise inserted into the multiple cloning site of pHR-SIN-BX-Strep-tag II. DNA encoding human 1G4 TCR α and β chains flanking a viral 2A peptide sequence was inserted into the multiple cloning site of pHR-SIN-BX-Twin-

Strep-tag such that the tag is at the N terminus of the TCRβ chain. 1G4 TCR was amplified without the native β chain signal peptide. The α and β chains also have a C-terminal FLAG tag and hemagglutinin (HA) tag, respectively. DNA encoding human 1G4 TCR α and β chains was a kind gift from Oreste Acuto, University of Oxford.

**Adaptor proteins.** DNA encoding human DNAX-activating protein of 12 kDa (DAP12) or FcRγ followed by C-terminal Myc tag was inserted into pHR-SIN-BX-IRES-EmGFP using 5′ BamHI and 3′ XhoI sites, retaining the plasmid IRES-EmGFP sequence.

**Ligand anchor.** Sequence: **METDTLLLWVLLLWVPGSTGD***YPYDVPDYA*TGGSAHI VMVDAYKPTK GGSGGS *HVSEDFTWEKPPEDPPDSKNTLVLFGAGFGAVITVVVIVVIIKC FCKHRSCFRRNEASRETNNSLTFGPEEALAEQTVFL*

DNA encoding the Igκ leader sequence (highlighted in bold), HA tag (in italics), SpyTag (underlined), and the extracellular hinge-like region, transmembrane, and intracellular regions of mouse CD80 (amino acids 227–306) (in italics and underlined) was inserted into pEE14 using 5′ HindIII and 3′ XbaI restriction sites: pEE14-ligand anchor.

**HLA-A\*02 single-chain dimer.** Sequence: *MSRSVALAVLAILSLSGLEAIQRTPKIQVYS RHPAENGKSNFLNCYVSGFHP SDIEVDLLKNGERIEKVEHSDLSFSKDWSFYLLYYTEFTPTE KDEYACRV NHVTLSQPKIVKWDRDM***GGGGSGGGGSGGGGS** GSHSMRYFFTSVSRPGR GEPRFIAVGYVDDTQFVRFDSDAASQRM EPRAPWIEQEGPEYWDGETRKVKAHSQT HRVDLGTLRGYYNQSEA GSHTVQRMYGCDVGSDWRFLRGYHQYAYDGKDYIALKED LRSWTA ADMAAQTTKHKWEAAHVAEQLRAYLEGTCVEWLRRYLENGKETLQ RTDA PKTHMTHHAVSDHEATLRCWALSFYPAEITLTWQRDGEDQTQ DTELVETRPAGDGT FQKWAAVVVPSGQEQRYTCHVQHEGLPKPLTL RWEPGSQPTIPIVGIIAGLVLFGAVIT GAVVAAVMWRRKSSDRKGGSY SQAASSDSAQGSDVSLTACKV

DNA encoding human β-2 microglobulin (in italics) and HLA-A*02 (amino acids 25–365) (underlined) separated by a GS linker ([GGGGS]$_3$) (in bold) was inserted into pDisplay using 5′ HindIII and 3′ XhoI restriction sites to create a single-chain dimer (SCD): pDISPLAY-SCD.

**Strep-Tactin, dead streptavidin-SpyCatcher, and variants.** Strep-Tactin sequence: MAEAGITGTWYNQLGSTFIVTAGADGALTGTYVTARGNAESRYVLTGR YDSAPATDG SGTALGWTVAWKNNYRNAHSATTWSGQYVGGAEARINTQ WLLTSGTTEANAWKST LVGHDTFTKVKPSAAS

Dead streptavidin-SpyCatcher sequence: MAEAGITGTWYAQLGDTFIVTAGADGALTG TYEAAVG **DDDGDDDGDDDG** AESRYVLTGRYDSAPATDGSGTALGWTVAWKNNYR NAHSATTWSGQYVGG AEARINTQWLLTSGTTEANAWKSTLVGHDTFTKVKPSAAS GSGSG *DYDIPTTENLYFQGAMVDTLSGLSSEQGQSGDMTIEEDSAT HIKFSKRDEDGKELA GATMELRDSSGKTISTWISDGQVKDFYL YPGKYTFVETAAPDGYEVATAITFTVNEQGQV TVNGKATKGDAHI*

Dead streptavidin is underlined, SpyCatcher is in italics, and the polyaspartate sequence is in bold.

Dead streptavidin sequence: MAEAGITGTWYAQLGDTFIVTAGADGALTGTYEAAVG NAESRYVLTGRY DSAPATDGSGTALGWTVAWKNNYRNAHSATTWSGQYVGGAEAR INTQW LLTSGTTEANAWKSTLVGHDTFTKVKPSAAS

Strep-Tactin-SpyCatcher Δ sequence: MAEAGITGTWYNQLGSTFIVTAGADGALTGTY VTARGNAESRYVLTGRYD SAPATDGSGTALGWTVAWKNNYRNAHSATTWSGQYVGG AEARINTQWLLT SGTTEANAWKSTLVGHDTFTKVKPSAAS **DDDGDDDGDDDD** *SAT HIKFSKRDEDGKELAGATMELRDSSGKTISTWISDGQVKDFYLYPGKYTF VETAAPDGYE VATAITFTVNEQGQVTVNGKATKGDAHI*

Strep-Tactin is underlined, SpyCatcherΔ is in italics, and the polyaspartate sequence is in bold. pET21–Strep-Tactin, pET21–dead streptavidin-SpyCatcher, and pET21–dead streptavidin (Addgene plasmid 20859) were gifts from Mark Howarth, University of Oxford [9].

Streptavidin sequence: MAEAGITGTWYNQLGSTFIVTAGADGALTGTYESAVGNAESR YVLTGRY DSAPATDGSGTALGWTVAWKNNYRNAHSATTWSGQYVGGAEARINTQW LLTSGTTEANAWKSTLVGHDTFTKVKPSAAS

The dead streptavidin sequence of pET21–dead streptavidin-SpyCatcher matches Addgene plasmid 59547 with polyaspartate sequence in the streptavidin 3/4 loop for anion exchange chromatography [10]. The C-terminal SpyCatcher sequence is different, however, as per Addgene plasmid 35044 [11]. pET21–Strep-Tactin-SpyCatcher Δ was created by amplification of Strep-Tactin and truncated SpyCatcher (as in Addgene plasmid 59547) sequences and insertion at either side of a polyaspartate sequence [10]. SpyCatcher truncation does not significantly affect SpyTag-SpyCatcher reaction efficiency [12].

**Nanobody CARs.** DNA encoding the Igκ leader sequence, LaG17 or LaM8 nanobody, IgG4 hinge region (amino acids 99–110 of IgG4 chain constant region), CD28 transmembrane region (amino acids 151–185), and ζ chain intracellular region (amino acids 52–164) was inserted into pHR-SIN-BX-IRES-EmGFP using 5′ BamHI and 3′ NotI sites replacing the IRES-EmGFP sequence [13].

**Fluorescent protein-SpyCatcherΔ fusion proteins.** DNA encoding GFP A206K or monomeric Cherry (mCherry), followed by SpyCatcherΔ, was inserted into pTrcHis using 5′ NheI and 3′ HindIII sites flanking the entire sequence.

## Cell lines

**THP-1, Jurkat, HEK293T cell lines.** THP-1, Jurkat, Jurkat reporter (enhanced green fluorescent protein [eGFP] production under the control of the nuclear factor kappa-light-chain-enhancer of activated B cells [NFκB] promoter), and human embryonic kidney 293T (HEK293T) cells were maintained in RPMI-1640 (Sigma-Aldrich R8758) media supplemented with 10% foetal bovine serum (FBS) and 100 U mL$^{-1}$ penicillin/streptomycin (Thermo Fisher Scientific 15140122) at 37˚C in a 5% CO$_2$-containing incubator. Jurkat NFκB-driven eGFP reporter and Jurkat NFκB-driven eGFP reporter 1G4 TCRα/β CD8α/β cells were a gift from Peter Steinberger and Wolfgang Paster, Medical University of Vienna [14, 15].

**CHO cell lines.** Chinese hamster ovary (CHO) mock cells were maintained in DMEM (Sigma-Aldrich D6429) supplemented with 5% FBS and 100 U mL$^{-1}$ penicillin/streptomycin. CHO ligand anchor cells were maintained in L-Glutamine-free DMEM (Sigma-Aldrich D6546) supplemented with 5% dialysed FBS (dialysed thrice against 10 L PBS), 100 U mL$^{-1}$ penicillin/ streptomycin, 1x GSEM supplement (Sigma-Aldrich G9785), and 50 μM L-Methionine sulfoximine (Sigma-Aldrich M5379). CHO ligand anchor HLA-A*02 cells were maintained in the above media additionally supplemented with 1 mg mL$^{-1}$ G-418 (Thermo Fisher Scientific 10131027). All CHO cell lines were maintained at 37˚C in a 10% CO$_2$-containing incubator.

## Lentiviral transduction of THP-1 and Jurkat cells

Receptor-expressing lentivector either alone or with the appropriate adaptor-expressing lentivector was cotransfected with the lentiviral packaging plasmids pRSV-Rev (Addgene plasmid 12253), pMDLg/pRRE (Addgene plasmid 12251), and pMD2.g (Addgene plasmid 12259) into HEK293T cells using X-tremeGENE 9 (Roche) as per the manufacturer's instructions [16]. Lentiviral packaging plasmids were a gift from Didier Trono, École polytechnique fédérale de Lausanne. Two days after transfection, viral supernatants were harvested, filtered, and used for the transduction of either THP-1 or Jurkat cells in the presence of 5μg mL$^{-1}$ Polybrene. For CD8α/ β-expressing Jurkat cells, Jurkat reporter cells expressing 1G4 TCRα/β tagged with Strep-tag II or Twin-Strep-tag were lentivirally transduced as above with pHR-SIN-BX-CD8b-T2A-CD8a (a kind gift of Peter Steinberger and Wolfgang Paster, Medical University of Vienna).

### Analysing receptor, adaptor, and coreceptor expression using flow cytometry and cell sorting by fluorescence-activated cell sorting (FACS)

Cells were analysed for receptor surface expression by flow cytometry using anti-Strep-tag II antibody Oyster 645 (IBA Lifesciences 2-1555-050) or anti-Strep-tag II antibody (IBA Lifesciences 2-1507-001) and anti-mouse IgG1 antibody Alexa Fluor 647 (Thermo Fisher Scientific A-21240) (BD FACSCalibur, 640-nm laser, FL4–661/16 band-pass filter, BD Biosciences). Introduced adaptor expression was tested via expression of EmGFP encoded on the adaptor lentivector (BD FACSCalibur, 488-nm laser, FL1–530/30 band-pass filter, BD Biosciences). Expression of 1G4 TCRβ and CD8α was analysed using anti-TCR Vβ 13.1 antibody FITC (H131; Thermo Fisher Scientific) and anti-CD8α antibody PE (HIT8a; Biolegend), respectively (BD FACSCalibur, 488-nm laser, FL1–530/30 band-pass filter and FL2–585/42 band-pass filter, BD Biosciences). Cells were sorted for high expression of receptor, introduced adaptor, or CD8 coreceptor by FACS (MoFlo Astrios, Beckman Coulter).

### pMHC tetramer staining of Jurkat NFκB eGFP cells

HLA-A*02:01 heavy chain (residues 1–278) with C-terminal BirA tag and β2-microglobulin were expressed in *Escherichia coli* as inclusion bodies, refolded in the presence of peptide (SLLMWITQV), and purified using size-exclusion chromatography [17]. NY-ESO-1 (157–165; SLLMWITQV) peptide was purchased at more than 95% purity (GenScript). Purified peptide-HLA-A*02 was biotinylated in vitro by BirA enzyme (Avidity) and mixed with Strep-tavidin:RPE (Bio-Rad STAR4A) to create tetramers. Cells were incubated with a below-saturating concentration of pMHC tetramers:RPE and analysed by flow cytometry (BD FACSCalibur, 488-nm laser, FL2–585/42 band-pass filter, BD Biosciences).

### Transfection of CHO cells with ligand anchor and HLA-A*02

CHO cells were transfected with either pEE14 (CHO mock) or pEE14-ligand anchor (CHO ligand anchor) using Xtreme-GENE 9 as per the manufacturer's instructions. A monoclonal population of CHO ligand anchor cells, created by limiting dilution, were transfected with pDISPLAY-SCD (CHO ligand anchor HLA-A*02) using Xtreme-GENE 9 as per the manufacturer's instructions. Transfected lines were cultured in the appropriate selection media after 48 hours.

### Checking ligand anchor and HLA-A*02 expression by flow cytometry and cell sorting by FACS

Cells were analysed for ligand anchor or HLA-A*02 surface expression by flow cytometry using anti-HA-Tag antibody Alexa Fluor 647 (6E2; Cell Signaling Technology) and anti-HLA-A2 FITC antibody (BB7.2; Santa Cruz Biotechnology), respectively (BD FACSCalibur, 640-nm laser, FL4–661/16 band-pass filter, 488-nm laser, FL1–530/30 band-pass filter, BD Biosciences). CHO ligand anchor HLA-A*02 cells were sorted for high expression of ligand anchor and HLA-A*02 using FACS (MoFlo Astrios, Beckman Coulter).

### Expression and purification of trivalent Strep-Tactin-SpyCatcher and monovalent Strep-Tactin-SpyCatcherΔ

Individual subunits (Strep-Tactin and dead streptavidin-SpyCatcher or dead streptavidin and Strep-Tactin-SpyCatcherΔ) were expressed in *E. coli* BL21-CodonPlus (DE3)-RIPL cells (Agilent Technologies 230280) and refolded from inclusion bodies using a modified version of the protocol previously described by Howarth and Ting [18]. Inclusion bodies were washed in

BugBuster (Merck Millipore 70921) supplemented with lysozyme, protease inhibitors, DNase I, and magnesium sulphate as per the manufacturers' instructions. Subunits were then mixed at a 3:1 molar ratio in order to bias refold towards the desired tetramer: Strep-Tactin and dead streptavidin-SpyCatcher to yield trivalent Strep-Tactin-SpyCatcher or dead streptavidin and Strep-Tactin-SpyCatcherΔ to yield monovalent Strep-Tactin-SpyCatcherΔ. Tetramers were refolded by rapid dilution and precipitated using ammonium sulphate precipitation. Precipitated protein was resuspended in 20 mM Tris (pH 8.0), filtered (0.22-μm filter), and loaded onto a Mono Q HR 5/5 column (GE Healthcare Life Sciences). Desired tetramers were eluted using a linear gradient of 0–0.5 M NaCl in 20 mM Tris (pH 8.0), concentrated, and buffer exchanged into 20 mM MES, 140 mM NaCl (pH 6.0).

## Expression and purification of GFP-SpyCatcherΔ and mCherry-SpyCatcherΔ

GFP-SpyCatcherΔ and mCherry-SpyCatcherΔ were expressed in *E. coli* BL21-CodonPlus (DE3)-RIPL cells (Agilent Technologies 230280). Bacterial cell pellets were washed in BugBuster (Merck Millipore 70921) supplemented with lysozyme, protease inhibitors, DNase I, and magnesium sulphate as per the manufacturers' instructions. The lysates were supplemented with 10 mM imidazole, mixed with Nickel-NTA agarose, and loaded onto gravity-flow columns. Loaded columns were washed with 50 mM $NaH_2PO_4$, 300 mM NaCl, 20 mM imidazole (pH 8), and GFP/mCherry-SpyCatcherΔ was eluted with 50 mM $NaH_2PO_4$, 300 mM NaCl, 250 mM imidazole (pH 8). Proteins were then concentrated and buffer exchanged into 20 mM MES, 140 mM NaCl (pH 6.0).

## SDS-PAGE and western blotting

Boiled samples were subjected to SDS-PAGE under reducing conditions on NuPAGE 4–12% Bis-Tris protein gels (Thermo Fisher Scientific NP0322) and stained using InstantBlue protein stain (Expedeon).

For western blotting, CHO ligand anchor cells presenting trivalent Strep-Tactin-SpyCatcher were lysed in Tris-buffered saline 1% NP40 at 4°C. Cleared and boiled cell lysates were then subjected to SDS-PAGE under reducing conditions on NuPAGE 4–12% Bis-Tris protein gels. Proteins were transferred to 0.2-μm PVDF membrane (GE Healthcare 10600022) using a semidry blotting system. The membrane was probed with anti-HA tag antibody (6E2; Cell Signaling Technology) followed by IRDye 680RD goat anti-mouse IgG (LI-COR Biosciences 925–68070) and with anti-streptavidin antibody (Abcam Ab6676) followed by anti-rabbit IgG Dylight 800 (Thermo Fisher Scientific SA5-10036).

Gels and blots were imaged using a LI-COR Odyssey Sa imaging system (LI-COR Biosciences) and images analysed using ImageJ software.

## Biotin-4-fluorescein quenching assay

Valency of trivalent and monovalent Strep-Tactin-SpyCatcher was confirmed using the quenching activity of biotin-4-fluorescein (Sigma-Aldrich B9431) when bound to Strep-Tactin [19, 20]. Trivalent or monovalent Strep-Tactin-SpyCatcher was incubated with a titration of biotin-4-fluorescein concentrations in black, opaque 96-well plates for 30 minutes at 25°C in PBS 1% BSA. Fluorescence was measured ($\lambda_{ex}$ 485 nm, $\lambda_{em}$ 520 nm) using a SpectraMax M5 plate reader (Molecular Devices). Fluorescence values were corrected for background fluorescence before analysis.

Negative control (buffer alone) data were fitted with the linear regression formula (Eq 1), where Y is fluorescence (arbitrary units [AU]), M is the gradient, X is the concentration of

biotin-4-fluorescein (M), and B is the $y$ intercept.

$$Y = M \cdot X + B \tag{1}$$

Sample data were fitted with segmental linear regression equation set (Eqs 2–5) in which X is the biotin-4-fluorescein concentration (M), Y is fluorescence (AU), X0 is the biotin-4-fluorescein concentration at which the line segments intersect (M), slope1 is the gradient of the first line segment, slope2 is the gradient of the second line segment, and intercept1 is the Y value at which the first line segment intersects the $y$ axis.

$$Y1 = intercept1 + slope1 \cdot X \tag{2}$$

$$Y \text{ at } X0 = slope1 \cdot X0 + intercept1 \tag{3}$$

$$Y2 = Y \text{ at } X0 + slope2 \cdot (X - X0) \tag{4}$$

$$Y = IF(X < X0, Y1, Y2) \tag{5}$$

Term X0 was converted into an estimate of number of biotin-binding sites per tetramer using the concentration of Strep-Tactin-SpyCatcher added. For this, we assume complete binding of biotin-4-fluorescein to protein.

## Surface plasmon resonance

Monomeric teal fluorescent protein (mTFP) with C-terminal His tag (6xHis) and either an N-terminal Strep-tag II or Twin-Strep-tag was expressed in *E. coli* and purified by nickel affinity chromatography [21].

Affinity measurements were made using a Biacore T200 or 3000 (GE Healthcare). All experiments were performed at 37˚C using a flow rate of 10 μL min$^{-1}$ in HBS-EP buffer (0.01 M HEPES buffer (pH 7.4), 0.15 M NaCl, 3 mM EDTA, 0.005% Surfactant P20). SpyTag-containing peptide (AHIVMVDAYKPTKGGSGGSHHHHHHHHHHHHH) (SpyTag is underlined), purchased at 95% purity (Peptide Protein Research), was immobilised to a sensor chip CM5 (GE Healthcare). Either trivalent Strep-Tactin-SpyCatcher or monovalent Strep-Tactin-SpyCatcherΔ was immobilised to the chip via the SpyTag peptide at various levels. Equilibrium binding was measured for graded concentrations of either Strep-tag II–mTFP or Twin-Strep-tag–mTFP. The K$_D$ (M) values were obtained by simultaneously fitting all the data for Twin-Strep-tag–mTFP binding trivalent Strep-Tactin-SpyCatcher (or all the data for Strep-tag II–mTFP binding monovalent Strep-Tactin-SpyCatcherΔ) with Eq 6 and constraining the fitting such that the K$_D$ value is shared between repeats. Y is the specific binding of injected mTFP fusion protein (response units [RU]), Bmax is the maximum specific binding (RU), X is the concentration of injected mTFP fusion protein (M), and h is the Hill slope.

$$Y = \frac{Bmax \cdot X^h}{K_D{}^h + X^h} \tag{6}$$

## Generating CHO ligand cells

CHO ligand anchor or CHO ligand anchor HLA-A$^*$02 cells (or CHO mock cells as a control) were incubated with various concentrations of trivalent Strep-Tactin-SpyCatcher or monovalent Strep-Tactin-SpyCatcherΔ in 20 mM MES, 140 mM NaCl (pH 6.0), 1% BSA for 10 minutes at 25˚C unless otherwise stated. Unbound Strep-Tactin-SpyCatcher was removed by washing thrice with PBS 1% BSA.

To generate CHO CAR ligand cells, CHO ligand anchor or CHO mock cells as a control were incubated with various concentrations of GFP-SpyCatcherΔ or mCherry-SpyCatcherΔ in 20 mM MES, 140 mM NaCl (pH 6.0), 1% BSA for 10 minutes at 25˚C. Unbound SpyCatcherΔ fusion protein was removed by washing thrice with PBS 1% BSA. Relative levels of GFP-Spy-CatcherΔ or mCherry-SpyCatcherΔ presented by the CHO CAR ligand cells was analysed by flow cytometry (BD LSRFortessa X-20, 488-nm laser, 530/30 band-pass filter and 561-nm laser, 610/20 band-pass filter, BD Biosciences). Background median fluorescence intensities (MFIs), measured from cells incubated with buffer alone instead of GFP/mCherry-Spy-CatcherΔ, were subtracted from all corresponding sample MFI values respectively. These values were then fitted with Eq 7, where Y is the MFI (AU), Bmax is the maximum specific GFP/mCherry-SpyCatcherΔ binding indicated by MFI (in AU), X is the concentration of GFP/mCherry-SpyCatcherΔ added (M), and $K_D$ we use as the concentration of GFP/mCherry-Spy-CatcherΔ that yields 50% maximal binding to CHO cells (M).

$$Y = \frac{Bmax \cdot X}{K_D + X} \qquad (7)$$

For instances in which CHO ligand anchor cells were incubated with both GFP-Spy-CatcherΔ and mCherry-SpyCatcherΔ, cells were first incubated with 0.13 μM GFP-Spy-CatcherΔ in 20 mM MES, 140 mM NaCl (pH 6.0) 1% BSA for 10 minutes at 25˚C. Unbound SpyCatcherΔ fusion protein was removed by washing thrice with PBS 1% BSA. Cells were then incubated with titrating concentrations of the second SpyCatcherΔ fusion protein as above before washing. Levels of GFP-SpyCatcherΔ and mCherry-SpyCatcherΔ presented by the CHO CAR ligand cells were then analysed by flow cytometry as above.

## Measuring generic ligand numbers per cell

To measure the number of generic ligands on a cell population preincubated with a single concentration of trivalent Strep-Tactin-SpyCatcher or monovalent Strep-Tactin-SpyCatcherΔ, the above biotin-4-fluorescein fluorescence-quenching assay was used. A titration of biotin-4-fluorescein was incubated with known numbers of cells preincubated with trivalent Strep-Tactin-SpyCatcher or monovalent Strep-Tactin-SpyCatcherΔ (or buffer alone as a control). The X0 term (calculated from the curve fit using Eqs 2–5) was converted to an estimate of average generic ligand number per cell using Eq 8, where L is the average number of ligands per cell, X0 is the saturation concentration of biotin-4-fluorescein extracted (M), V is the sample volume (L), $N_A$ is Avogadro's constant, C is the number of cells in the sample, and B is the number of biotin-binding sites per ligand. In the case of trivalent Strep-Tactin-SpyCatcher and monovalent Strep-Tactin-SpyCatcherΔ, B = 3 or 1, respectively.

$$L = \frac{X0 \cdot V \cdot N_A}{C} / B \qquad (8)$$

To indicate relative levels of generic ligand per cell, CHO cells preincubated with trivalent Strep-Tactin-SpyCatcher or buffer alone were incubated for 30 minutes at 25˚C with 2 μM either ATTO 647 biotin (ATTO Technology AD 647–71) or ATTO 488 biotin (ATTO Technology AD 488–71), premixed with 6 μM biotin (Sigma-Aldrich B4501). The presence of biotin minimises the self-quenching activity of ATTO dye biotin conjugates. When monovalent Strep-Tactin-SpyCatcherΔ was used, cells were incubated with 2 μM ATTO 488 biotin (ATTO Technology AD 488–71) alone and treated as above. Cells were analysed by flow cytometry (BD FACSCalibur, 640-nm laser, FL4–661/16 band-pass filter, 488-nm laser, FL1–530/30 band-pass filter, BD Biosciences). Background MFIs (or geometric mean fluorescence

intensities [gMFIs] where stated), in which cells were incubated with buffer alone instead of Strep-Tactin-SpyCatcher, were subtracted from all corresponding sample MFI/gMFI values respectively. These values were then fitted with Eq 9, where Y is the MFI or gMFI (AU), Bmax is the maximum specific trivalent or monovalent Strep-Tactin-SpyCatcher binding indicated by MFI or gMFI respectively (in AU), X is the concentration of trivalent or monovalent Strep-Tactin-SpyCatcher added (μM), and $K_D$ we use as the concentration of Strep-Tactin-Spy-Catcher that yields 50% maximal binding to CHO cells (M). Whether the median or geometric mean was extracted from flow cytometry analyses is indicated on each graph.

$$Y = \frac{Bmax \cdot X}{K_D + X} \tag{9}$$

To convert Y values into estimates of average generic ligand number per cell, the ligand number per cell at saturating trivalent or monovalent Strep-Tactin-SpyCatcher concentration calculated from Eq 8 was substituted into Eq 9 as Bmax. Y values were then recalculated following this adjustment. Average ligand density (molecules/μm$^2$) was calculated from these estimates by assuming a CHO cell surface area of 700 μm$^2$. The latter is based on a diameter of 15 μm and an assumed spherical shape [22, 23].

We have independently verified the numbers of generic ligands per cell quantified via this method by flow cytometry using antibodies and IgG quantitation beads (S7 Fig).

## Measuring generic ligand numbers per cell using anti-mouse IgG quantitation beads

Anti-streptavidin antibody binds poorly to Strep-Tactin, and therefore, we used CHO ligand anchor cells incubated with trivalent streptavidin-SpyCatcher to compare ligand numbers calculated using a biotin-4-fluorescein fluorescence-quenching assay and quantitation beads (S7 Fig). Using a biotin-4-fluorescein fluorescence-quenching assay, we showed that trivalent Strep-Tactin and streptavidin yield similar numbers of ligand per cell (S7 Fig). Trivalent streptavidin-SpyCatcher was synthesised as described for trivalent Strep-Tactin-SpyCatcher.

Either anti-mouse IgG beads (Quantum Simply Cellular anti-Mouse IgG; Bangs Laboratories, 815) or CHO ligand anchor cells preincubated with trivalent Strep-Tactin-SpyCatcher or trivalent streptavidin-SpyCatcher were incubated with anti-streptavidin antibody PE (3A20.2; BioLegend). Cells and beads were analysed by flow cytometry (BD FACSCalibur, 488-nm laser, FL2–585/42 band-pass filter, BD Biosciences). A standard curve from the bead MFI values was created as per the manufacturer's instructions. Background MFI, in which cells were incubated with buffer alone, was subtracted from the sample value. This was used to interpolate the number of ligands per cell using the anti-mouse IgG bead standard curve.

For CHO ligand anchor cells incubated with 0.05 μM trivalent streptavidin-SpyCatcher, 110,000 ligands per cell was calculated using a biotin-4-fluorescein fluorescence-quenching assay, and 320,000 ligands per cell was calculated using anti-streptavidin antibody and anti-mouse IgG beads (S7 Fig).

## Down-regulation of generic ligand over time

CHO ligand anchor cells were incubated with 15 μM trivalent Strep-Tactin-SpyCatcher (or buffer alone) as described above and incubated at 37°C as for stimulation assays for the time points indicated. Cells were analysed for generic ligand surface expression using ATTO 647 biotin as above and normalised to the MFI value at time 0. To calculate the decay, the MFI values were fitted with Eq 10, where Y0 is the Y value when X = 0, Plateau is the Y value at which

the curve reaches a plateau, X is time in minutes, and K is the rate constant in inverse minutes.

$$Y = (Y0 \ - \ \text{Plateau}) \cdot e^{-K \cdot X} + \text{Plateau} \qquad (10)$$

### Measuring number of 9V-HLA-A*02 per cell

The α and β subunits of c58c61 high-affinity 1G4 TCR were expressed in *E. coli* as inclusion bodies, refolded in vitro, and purified using size-exclusion chromatography as described previously [24, 25]. This protein was then fluorescently labelled with Alexa Fluor 647 NHS Ester (Thermo Fisher Scientific A37573), and the degree of protein labelling was calculated as per the manufacturer's instructions. We will refer to this labelled high-affinity TCR as 1G4hi TCR AF647.

CHO mock or CHO ligand anchor HLA-A*02 cells were incubated with a titration of NY-ESO-1 9V peptide for 1–3 hours at 37°C as per stimulation assays. Washed cells were incubated with an above saturation concentration of 1G4hi TCR AF647 for 1 hour at 4°C. Cells were analysed by flow cytometry alongside Alexa Fluor 647 fluorescence quantitation beads (Bangs Laboratories 647) (BD FACSCalibur, 640-nm laser, FL4–661/16 band-pass filter, BD Biosciences). Bead MFI values were extracted and used to form a standard curve from which the Jurkat NFκB eGFP cell–specific binding MFIs were interpolated to give estimates of number of 9V-HLA-A*02 per cell, correcting for the degree of labelling of 1G4hi TCR AF647. In each experiment, the number of 9V-HLA-A*02 per cell for the top two or three cell subpopulations were extrapolated from the standard curve instead of interpolated.

### Cellular functional assays

During all functional assays, samples of prepared CHO cells were used for estimating the number of generic ligands or 9V-HLA-A*02 per cell, or the relative levels of GFP-SpyCatcherΔ or mCherry-SpyCatcherΔ presented by CHO ligand anchor cells.

**THP-1 cellular assays.** CHO mock or CHO ligand anchor cells ($2 \times 10^5$) preincubated with trivalent Strep-Tactin-SpyCatcher or buffer alone were mixed with Twin-Strep-tagged receptor- and adaptor-expressing THP-1 or untransduced THP-1 cells ($1 \times 10^5$) in DMEM 5% FBS, 100 U mL$^{-1}$ penicillin/streptomycin, 2 μg mL$^{-1}$ avidin. Cells were incubated in a 37°C 10% $CO_2$-containing incubator for 20 hours. Supernatants were harvested and assayed for interleukin-8 (IL-8) by ELISA (Thermo Fisher Scientific 88-8086-77). Alternatively, CHO ligand anchor cells ($2 \times 10^5$) preincubated with monovalent Strep-Tactin-SpyCatcherΔ, trivalent Strep-Tactin-SpyCatcher, or buffer alone were mixed with THP-1 Siglec-14-Strep-tag II DAP12 cells ($1 \times 10^5$) in DMEM 5% FBS, 100 U mL$^{-1}$ penicillin/streptomycin, 2 μg mL$^{-1}$ avidin. Cells were then incubated, supernatant harvested, and IL-8 secretion assayed as above.

**Jurkat NFκB eGFP cellular assays.** CHO mock or CHO ligand anchor HLA-A*02 cells ($2 \times 10^5$) preincubated with trivalent Strep-Tactin-SpyCatcher or buffer alone were mixed with untransduced Jurkat NFκB eGFP cells or Jurkat NFκB eGFP 1G4 TCRα/β-Twin-Strep-tag cells ($1 \times 10^5$) in DMEM 5% FBS, 100 U mL$^{-1}$ penicillin/streptomycin, 2 μg mL$^{-1}$ avidin. Alternatively, CHO mock or CHO ligand anchor HLA-A*02 cells ($2 \times 10^5$) were mixed with untransduced Jurkat NFκB eGFP cells or Jurkat NFκB eGFP 1G4 TCRα/β-Twin-Strep-tag cells ($1 \times 10^5$), and NY-ESO-1 9V peptide in DMEM 5% FBS, 100 U mL$^{-1}$ penicillin/streptomycin, 2 μg mL$^{-1}$ avidin. Cells were incubated in a 37°C 10% $CO_2$ incubator for 20 hours. Cells were harvested, incubated with anti-CD45 antibody Alexa Fluor 647 (F10-89-4; Bio-Rad Laboratories), and analysed by flow cytometry (BD FACSCalibur, 640-nm laser, FL4–661/16 band-pass filter; 488-nm laser, FL1–530/30 band-pass filter, BD Biosciences).

**Jurkat CAR cellular assays.** CHO mock or CHO ligand anchor cells were stained with Tag-it Violet cell tracking dye (BioLegend 425101). These cells were then incubated with GFP-SpyCatcherΔ, mCherry-SpyCatcherΔ, or buffer alone before being mixed with untransduced Jurkat cells, Jurkat LaG17-ζ, or Jurkat LaM8-ζ cells ($2 \times 10^5$ CHO cells to $1 \times 10^5$ Jurkat cells) in RPMI 10% FBS, 100 U mL$^{-1}$ penicillin/streptomycin. Cells were incubated in a 37˚C 5% $CO_2$ incubator for 20 hours. Cells were harvested, incubated with anti-CD69 antibody APC-Cy7 (FN50; BD Biosciences), and analysed by flow cytometry (BD LSRFortessa X-20, 405-nm laser, 450/50 band-pass filter and 640 nm laser, 780/60 band-pass filter, BD Biosciences).

**Analysis.** For THP-1 cell assays, IL-8 concentrations in negative controls (in which CHO cells were preincubated with buffer alone instead of Strep-Tactin-SpyCatcher) were subtracted from corresponding sample IL-8 concentrations to correct for background levels. Dose-response curves were then fitted with Eq 11, where Y is the measured cell response (pg mL$^{-1}$); Bottom and Top are the minimum and maximum cell response, respectively (pg mL$^{-1}$); $EC_{50}$ is the ligand number per cell that yields a half-maximal response; X is the number of generic ligands per cell; and Hill slope relates to the steepness of the curve.

$$Y = \text{Bottom} + \frac{\text{Top} - \text{Bottom}}{1 + 10^{(\text{Log}(EC_{50}-X))\cdot\text{Hillslope}}} \tag{11}$$

For Jurkat NF$\kappa$B eGFP cell assays, only CD45+ cells were analysed for eGFP. Negative control populations were used to set a gate on eGFP expression. The percentages of CD45+ cells positive for eGFP were extracted and corrected for corresponding background percentage of eGFP+ Jurkat cells (samples in which CHO cells were incubated with buffer alone instead of Strep-Tactin or peptide). Dose-response curves were fitted with Eq 11, where Y is the measured cell response (% or AU); Bottom and Top are the minimum and maximum cell response, respectively (% or AU); $EC_{50}$ is the ligand number per cell that yields a half-maximal response; X is the number of generic ligands or 9V-HLA-A*02 per cell; and Hill slope relates to the steepness of the curve. When comparing generic ligand and native ligand dose-response curves, the Y values were normalised to the individual data set maximal response, giving the maximum a value of 1.

For Jurkat CAR cell assays, only Tag-it Violet negative cells were analysed for CD69 expression. The CD69 expression MFI values of Tag-it Violet negative cells were extracted and corrected for the resting CD69 expression MFI (samples in which CHO cells were incubated with buffer alone instead of GFP/mCherry-SpyCatcherΔ). Dose-response curves were fitted with Eq 12, where Y is the CD69 MFI value above background (AU), and Bottom and Top are the minimum and maximum cell response, respectively (AU). X is the relative levels of GFP/mCherry-SpyCatcherΔ on the CHO cells, interpolated from Eq 7 fitted to the data within each experiment (AU), and $EC_{50}$ is the interpolated MFI of GFP/mCherry-SpyCatcherΔ presented on CHO cells that yields a half-maximal response.

$$Y = \text{Bottom} + \frac{\text{Top} - \text{Bottom}}{1 + 10^{(\text{Log}(EC_{50}-X))}} \tag{12}$$

# Results

## Design and development of a generic ligand system

We aimed to design a system requiring minimal manipulation of the receptor, in order to preserve its structure and interactions. In addition, we sought to create a cell surface–expressed ligand that would engage receptors with an affinity comparable to physiological NTR–ligand

interactions and that, when bound to receptor, would preserve the cell–cell intermembrane distance.

We developed a generic ligand based on the interaction between the peptide Twin-Strep-tag (sequence <u>WSHPQFEK</u>GGGSGGGSGGSA<u>WSHPQFEK</u>), which has two Strep-tag II motifs, and Strep-Tactin, a variant of streptavidin (outlined in Fig 1) [26–28].

The NTR of interest, expressed on a 'receptor' cell, is genetically engineered to add the Twin-Strep-tag peptide to the extracellular N terminus, where it is accessible for engagement by the generic ligand presented on another 'ligand' cell (Fig 1). This ligand is made up of two components: a cell surface ligand anchor and a soluble fusion protein that spontaneously forms a covalent bond with the anchor. The ligand anchor comprises the transmembrane and cytoplasmic portions of mouse CD80 fused to an N-terminal SpyTag peptide, forming one-half of the covalent bond-forming split-protein pair SpyTag/SpyCatcher [11, 29, 30]. The soluble fusion protein comprises trivalent Strep-Tactin, which has three binding sites for Strep-tag II motifs, fused to SpyCatcher (trivalent Strep-Tactin-SpyCatcher) [26, 27]. When soluble trivalent Strep-Tactin-SpyCatcher is incubated with cells expressing the ligand anchor, SpyTag and SpyCatcher spontaneously form a covalent bond. This yields a cell surface ligand able to bind a Twin-Strep-tagged receptor (Fig 1).

Based on the available structures of Strep-Tactin and SpyTag/SpyCatcher and estimates of linker lengths, we predict the complete generic ligand to have an extracellular length similar to 2–3 immunoglobulin-like domains when extended (Protein Data Bank [PDB] accession numbers: 1KL3, 4MLI) [12, 31]. This is comparable to physiological NTR ligands (discussed by Dushek and colleagues [1]).

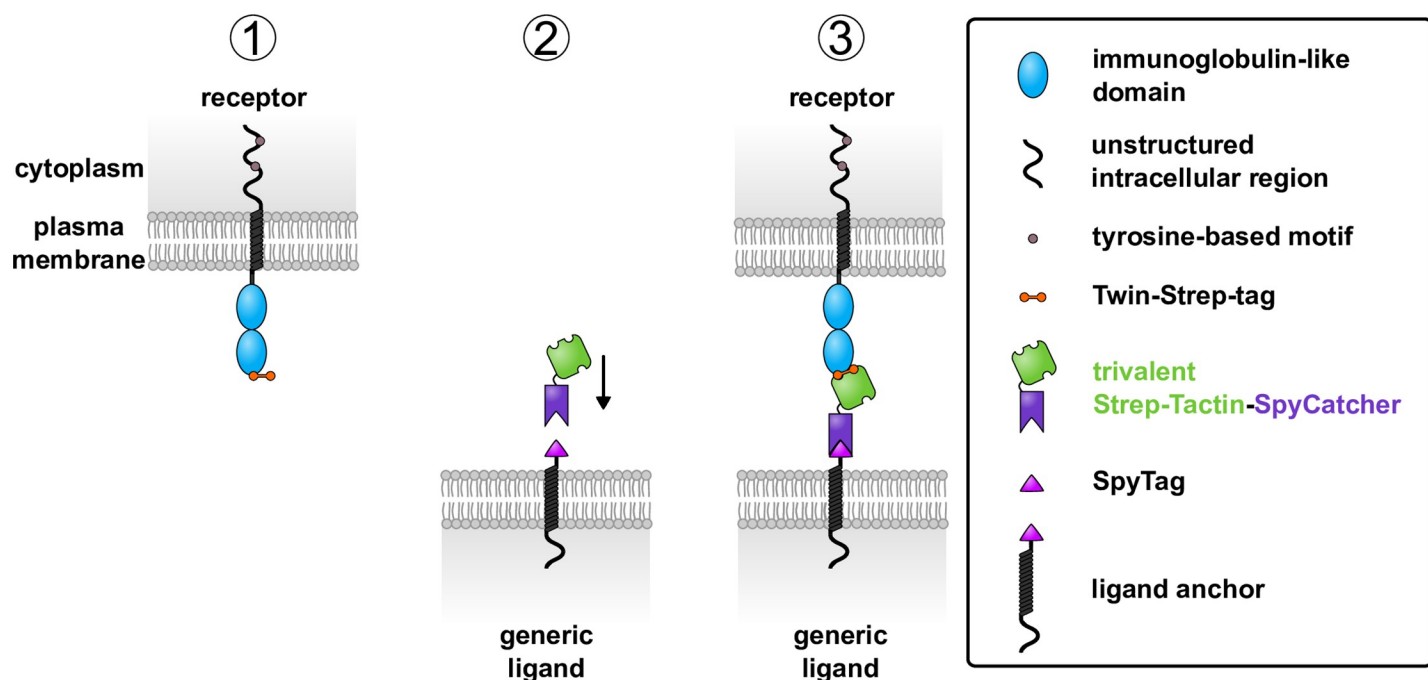

**Fig 1. Design of a generic ligand system.** (1) Each receptor is constructed with a Twin-Strep-tag at the extracellular terminus and expressed in cell lines. (2) The generic ligand is made of two components: a cell surface–expressed ligand anchor with N-terminal SpyTag and soluble trivalent Strep-Tactin-SpyCatcher protein. Trivalent Strep-Tactin-SpyCatcher is added to cells expressing the generic ligand anchor. When SpyCatcher and SpyTag interact, a spontaneous covalent isopeptide bond forms between them, creating the complete generic ligand. (3) The three binding sites of trivalent Strep-Tactin-SpyCatcher are available for ligation by the Twin-Strep-tagged receptor. Two binding sites are required for the full engagement of Twin-Strep-tag.

## Preparation of trivalent Strep-Tactin-SpyCatcher

To prepare trivalent Strep-Tactin-SpyCatcher tetramers, we employed a method previously used to generate streptavidin tetramers of defined valency [9, 32]. This uses a mutated streptavidin subunit that has negligible biotin-binding activity, termed 'dead' streptavidin. Biotin and Strep-tag II occupy the same surface pocket of streptavidin, and so we assumed that the dead streptavidin subunit is also unable to bind Strep-tag II [33]. Strep-Tactin subunits were refolded from bacterial inclusion bodies with subunits of dead streptavidin fused at its C terminus to SpyCatcher (dead streptavidin-SpyCatcher) in a 3:1 molar ratio (S1 Fig).

Dead streptavidin-SpyCatcher contains a polyaspartate insertion allowing purification of the trivalent Strep-Tactin-SpyCatcher tetramer from other possible configurations using anion exchange chromatography (S1 Fig).

We analysed a sample from the first eluted peak, predicted to be that of trivalent Strep-Tactin-SpyCatcher, using SDS-PAGE. Upon boiling, the tetramer is reduced to individual monomers of Strep-Tactin and dead streptavidin-SpyCatcher, allowing visualisation of the relative proportion of the subunits (S1 Fig). For comparison, we analysed a sample from a later peak, which we predict to contain monovalent Strep-Tactin (1 × Strep-Tactin, 3 × dead streptavidin-SpyCatcher) based on order of elution. The ratio of Strep-Tactin to dead streptavidin-SpyCatcher subunits was higher than expected (4.7:1 for trivalent Strep-Tactin-SpyCatcher) but was consistent with a 3:1 ratio when compared to the subunit ratio of the monovalent protein (S1 Fig).

To confirm that the purified protein was the desired tetramer, we used a biotin-4-fluorescein fluorescence-quenching assay previously used to estimate the number of biotin-binding sites per streptavidin tetramer (S1 Fig) [32]. When bound to Strep-Tactin, the fluorescence of biotin-4-fluorescein is quenched. As the concentration of biotin-4-fluorescein added to trivalent Strep-Tactin-SpyCatcher increases above binding-site saturation, there is an increasing amount of free (nonquenched) biotin-4-fluorescein in solution. This is visualised as a sharp increase in fluorescence, with the inflection point indicating saturation. Titration of biotin-4-fluorescein yielded an estimated number of 4.2 binding sites per trivalent Strep-Tactin-SpyCatcher (S1 Fig). Although higher than the expected value of 3, it is three times the estimated binding-site number calculated for the predicted monovalent Strep-Tactin peak (1.4). We assume there are the same number of available Strep-tag II–binding sites per tetramer.

## Characterisation of the generic ligand

**Affinity between Twin-Strep-tag and trivalent Strep-Tactin-SpyCatcher.** We used surface plasmon resonance and a Twin-Strep-tag–mTFP fusion protein to analyse Twin-Strep-tag binding to trivalent Strep-Tactin-SpyCatcher at 37°C. The $K_D$ was measured as 6.8 μM (Fig 2A). This is comparable to the affinities reported for physiological NTR–ligand interactions we wish to replicate [34–36].

**Characterising the optimal conditions for ligand anchor:trivalent Strep-Tactin-SpyCatcher binding.** A stable, high-expressing ligand anchor cell line was established and maintained under selection (S2 Fig). CHO cells were used to avoid any confounding receptor–ligand interactions that might occur if both receptor and ligand cells were human.

We explored the optimal conditions for covalent coupling between the cell surface–presented ligand anchor and soluble trivalent Strep-Tactin-SpyCatcher. In line with the findings of Zakeri and colleagues, we found that coupling between CHO ligand anchor cells and trivalent Strep-Tactin-SpyCatcher was most efficient in buffer at pH 5–6 (S2 Fig) [11]. The widest range of surface densities was achieved with a 5–10-minute incubation of trivalent Strep-Tactin-SpyCatcher at a wide range of concentrations (S2 Fig).

Using western blotting on boiled cell lysates separated by SDS-PAGE, we visualised the ligand anchor by probing for the N-terminal HA tag (S2 Fig). Addition of trivalent Strep-Tactin-SpyCatcher to the cells led to a substantial increase in the molecular weight of a significant proportion of ligand anchor consistent with covalent coupling to the soluble fusion protein. Probing with anti-streptavidin antibody confirmed this (S2 Fig).

A time course showed that a significant proportion of generic ligand remains at the cell surface for many hours post-reconstitution (S2 Fig). Receptor stimulation assays are commonly conducted over this time frame so that the ligand is able to provide a strong stimulus for this duration.

## Measuring the number of generic ligands per cell

A major strength of the generic ligand system is that the ligand dose can be varied easily by titrating the concentration of soluble trivalent Strep-Tactin-SpyCatcher added to cells. To enable us to determine the ligand surface density required for activation, we developed a method to measure the number of generic ligands per cell. This method uses two assays. The first, whereby CHO generic ligand cells are incubated with ATTO 647 biotin, gives an indication of how the relative number of generic ligand sites varies with Strep-Tactin-SpyCatcher concentration (Fig 2B).

The second assay measures the average maximum number of generic ligands per cell in a population of cells saturated with trivalent Strep-Tactin-SpyCatcher. This uses the same biotin-4-fluorescein fluorescence-quenching assay shown in S1 Fig. Assuming complete binding, the biotin-4-fluorescein concentration that saturates the cells is indicated by the inflection point of the graph (Fig 2C). This saturating concentration for a known number of cells in a defined volume can then be converted into an average number of generic ligands per cell (see the Measuring generic ligand numbers per cell section). By combining the absolute number of generic ligands per cell at saturation (Fig 2C) and the relative ligand levels across a range of soluble trivalent Strep-Tactin-SpyCatcher concentrations (Fig 2B), the average number of generic ligands per cell for a given soluble trivalent Strep-Tactin-SpyCatcher concentration can be estimated (Fig 2D). A maximum of 3 million generic ligands per cell can be consistently achieved, and the ligand dose can therefore easily be varied over several orders of magnitude.

## Representative activating NTRs can be stimulated by generic ligand

Representative activating human NTRs from various receptor families were genetically modified to have an N-terminal Twin-Strep-tag (Fig 3A).

SIRPβI (CD172b) and Siglec-14 are expressed in a number of leukocytes, including monocytes and macrophages [37, 38]. Siglec-14 binds to sialic acid presented by numerous bacteria to induce responses including cytokine secretion [38]. There is evidence that SIRPβI contributes to neutrophil transepithelial migration and macrophage phagocytosis, but a ligand has yet to be identified [37]. A generic ligand is therefore very useful for the study of SIRPβI. As a member of the NK cell cytotoxicity family, NKp30 (CD337) is expressed in NK cells and binds both pathogen and cellular ligands to mediate NK cell cytotoxicity [39].

SIRPβI, Siglec-14, and NKp30 receptors each associate with adaptor proteins that contain cytoplasmic phosphorylatable tyrosine residues that mediate immune signalling [37–39]. Therefore, to establish stable cell lines, THP-1 cells were cotransduced with lentiviruses encoding the tagged receptor and appropriate adaptor (S3 Fig).

The αβ TCR complex consists of TCR α and β chains associated with a TCRζ homodimer and CD3δε and CD3γε heterodimers. The 1G4 TCR α chain and Twin-Strep-tagged β chain were transduced into a Jurkat nuclear factor kappa-light-chain-enhancer of activated B cells

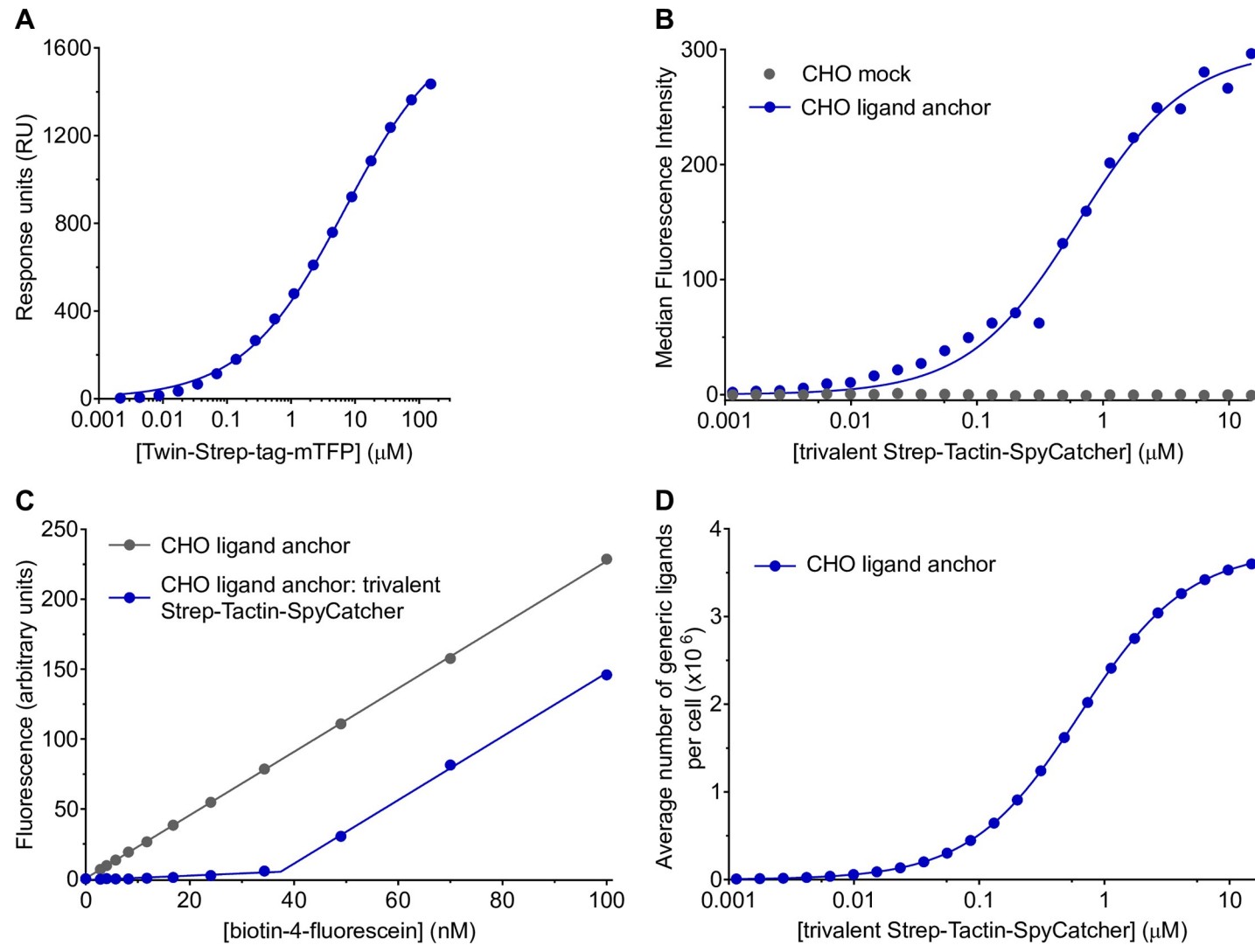

**Fig 2. Characterisation of Twin-Strep-tag:trivalent Strep-Tactin-SpyCatcher interaction and quantifying the number of generic ligands per cell.** (A) Representative equilibrium binding measured by surface plasmon resonance of Twin-Strep-tag–mTFP injected over immobilised trivalent Strep-Tactin-Spycatcher at 37°C is shown. The $K_D$ (SEM) for the collated data ($n = 11$) is 6.8 μM (0.62 μM), and the mean Hill slope (SEM) is 0.46 (0.03) to 2 s.f. (B) A relative indication of the level of generic ligand per cell as a function of trivalent Strep-Tactin-SpyCatcher concentration added to cells. Median fluorescence intensity values extracted from flow cytometry analyses of cells incubated with ATTO 647 biotin are shown. (C) CHO ligand anchor cells preincubated with trivalent Strep-Tactin-SpyCatcher or buffer alone were incubated with a titration of biotin-4-fluorescein in a fluorescence-quenching assay. The inflection point is used to calculate average absolute number of generic ligands per cell. (D) The saturating concentration of biotin-4-fluorescein was extracted from C and converted into number of generic ligands per cell. This was substituted as the maximum into the fitted curve in (B) to interpolate the average number of generic ligands per cell as a function of trivalent Strep-Tactin-SpyCatcher concentration added to cells. Summary numerical data are provided in S1 Data. CHO, Chinese hamster ovary; mTFP, monomeric teal fluorescent protein; SEM, standard error of the mean; s.f., significant figures.

(NFκB) reporter cell line in which the production of eGFP is under the control of the NFκB promoter (S3 Fig) [14, 15]. Any 1G4 TCR α and β chains expressed at the cell surface are presumed to be associated with endogenous TCR/CD3 signalling subunits.

All four tagged receptors were activated by generic ligand cells with a clear dose-dependent response, which increased with the number of generic ligands per cell (Fig 3B–3E). This response was specific as shown by the absence of IL-8 secretion or NFκB-driven eGFP expression in samples containing negative control receptor cells (Fig 3B–3E). The EC$_{50}$ values of the receptor responses from three independent experiments are shown in Fig 3F. Thus, the generic

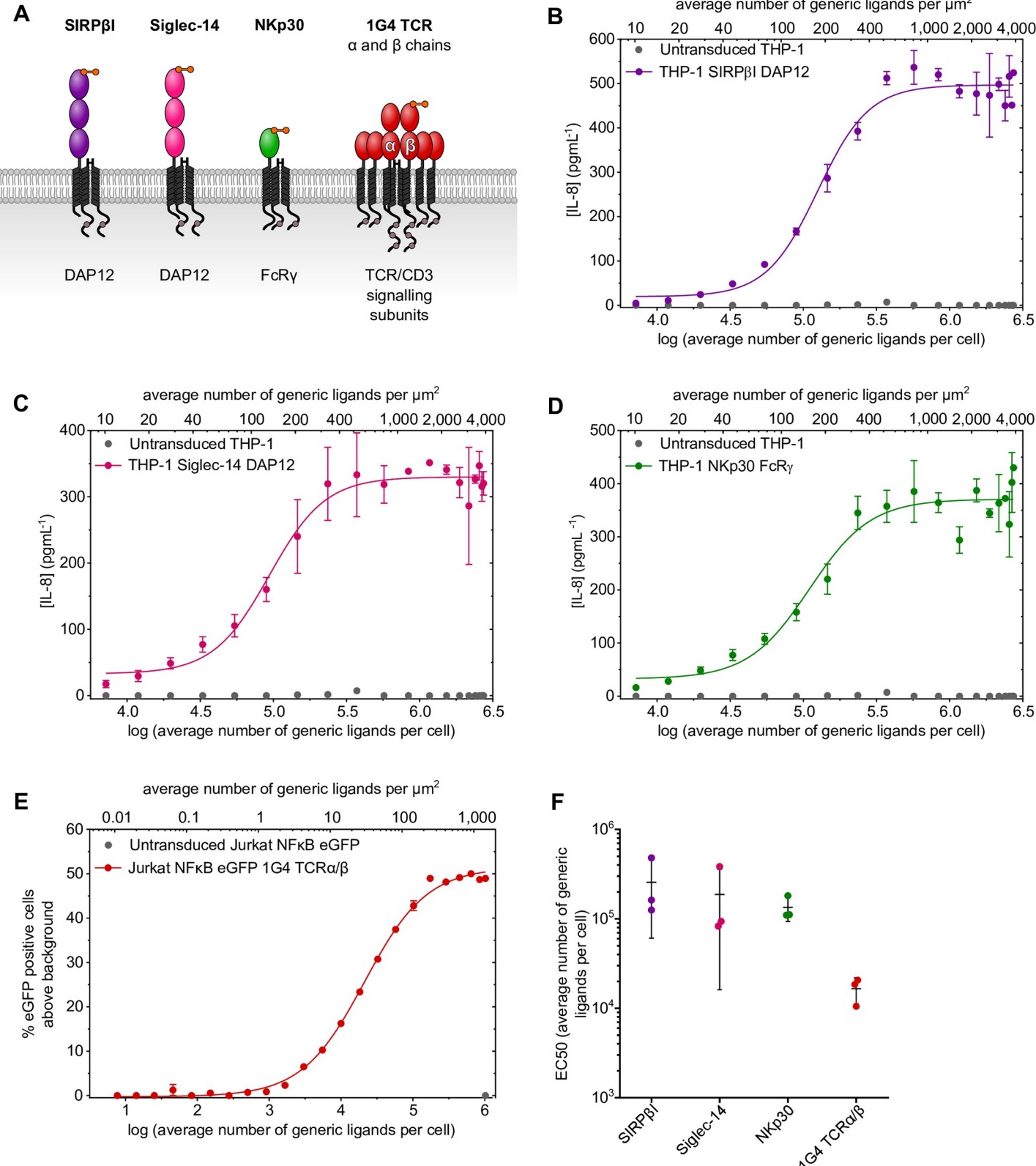

**Fig 3. Twin-Strep-tagged receptors can be stimulated by generic ligand-presenting cells.** (A) Cartoon depictions of four representative NTRs with Twin-Strep-tags and adaptor proteins are shown. In the case of 1G4 TCR, the β chain has an N-terminal Twin-Strep-tag. The extracellular region of each receptor contains one or more

immunoglobulin-like domains (see legend in Fig 1). Response of Twin-Strep-tagged SIRPβI (B), Siglec-14 (C), or NKp30 (D) expressing THP-1 cells to generic ligand presented on CHO cells. Receptor response is measured by IL-8 secretion. (E) Response of Jurkat NFκB eGFP 1G4 TCRα/β-Twin-Strep-tag cells to generic ligand presented on CHO cells. Receptor response, indicated by eGFP expression under the control of the NFκB promoter, is shown as percentage of cells positive for eGFP above background. Error bars indicate the range ($n = 2$), and data are representative of three independent experiments. Within each stimulation, a sample of CHO cells were taken and used to measure the number of generic ligands per cell as in (Fig 2). Ligand density was calculated from these numbers using an estimated CHO cell surface area of 700 $\mu m^2$ (see the Measuring generic ligand numbers per cell section) [22, 23]. (F) EC50 values from individual experiments of THP-1 SIRPβI DAP12, THP-1 Siglec-14 DAP12, THP-1 NKp30 FcRγ, or Jurkat NFκB eGFP 1G4 TCRα/β cells responding to generic ligand are plotted. Bars indicate the mean and standard deviation ($n = 3$). Summary numerical data are provided in S1 Data. CHO, Chinese hamster ovary; DAP12, DNAX-activating protein of 12 kDa; eGFP, enhanced green fluorescent protein; IL-8, interleukin 8; NFκB, nuclear factor kappa-light-chain-enhancer of activated B cells; NK, natural killer; NTR, non-catalytic tyrosine-phosphorylated receptor; Siglec-14, Sialic acid–binding immunoglobulin-type lectin 14; SIRPβI, signal regulatory protein βI; TCR, T-cell receptor.

ligand can bind to and trigger several different receptors bearing an N-terminal Twin-Strep-tag.

## Twin-Strep-tagged TCR responds to generic ligand and native ligand with a similar sensitivity

In order to validate this approach, we compared the response of the TCR to generic ligand with its response to native ligand, pMHC.

1G4 TCR and its cognate ligand NY-ESO-1$_{157–165}$ 9V peptide variant (SLLMWITQV) presented in complex with HLA-A*02 is a well-characterised receptor–ligand pair [40–42]. The dissociation constant of 1G4 TCR binding to 9V-HLA-A*02 ($K_D$ = 6–7 μM) is comparable to the $K_D$ we have measured for Twin-Strep-tag and trivalent Strep-Tactin-SpyCatcher (Fig 2A) [41, 42].

To compare the TCR response to either generic or native ligand, the Jurkat NFκB eGFP reporter cell line transduced with 1G4 TCR α and Twin-Strep-tagged β chains (S3 Fig) was presented to CHO cells that express both HLA-A*02 in the form of an SCD and the generic ligand anchor (Fig 4A, S3 Fig). These CHO cells were either preincubated with trivalent Strep-Tactin-SpyCatcher or loaded with 9V peptide. To allow a direct comparison between generic and pMHC ligand, the Jurkat NFκB eGFP cells were not transduced with the coreceptor CD8, since it binds MHC but not generic ligand.

In order to quantitatively compare the TCR response to generic ligand or 9V-HLA-A*02, we measured the number of 9V-HLA-A*02 molecules on the CHO ligand cells. For this, we used a soluble affinity-enhanced (c58/c61) form of the 1G4 TCR that binds to 9V-HLA-A*02 with a much higher affinity than the wild-type TCR ($K_D$ = 71 pM) [25, 43]. This soluble 1G4 high-affinity TCR was labelled with Alexa Fluor 647 (1G4hi TCR AF647) and used in combination with fluorescence quantitation beads to interpolate the average number of 9V-HLA-A*02 per cell (Fig 4B, S4 Fig). Incubating CHO ligand anchor HLA-A*02 cells with varying concentrations of 9V peptide yielded a large dynamic range of ligand number per cell (Fig 4B). Therefore, we were able to present Jurkat NFκB eGFP 1G4 TCRα/β-Twin-Strep-tag cells with CHO ligand anchor HLA-A*02 cells presenting either 9V-HLA-A*02 or generic ligand at similar densities.

Jurkat NFκB eGFP 1G4 TCRα/β-Twin-Strep-tag cells responded to both 9V-HLA-A*02 and generic ligand in a dose-dependent, specific manner, visualised using the NFκB reporter (Fig 4C). The EC50 values from three independent experiments are shown in Fig 4D. There is a 2-fold difference in the average sensitivity between the receptor response to 9V-HLA-A*02 versus generic ligand.

Based on the structure of soluble 1G4 TCRα/β bound to cognate pMHC, we predict the presence of Twin-Strep-tag should not interfere with TCR-pMHC binding (PDB accession number: 2BNR) [41]. We used soluble pMHC class I tetramer staining to confirm this. Because tetramer staining of Jurkat NFκB eGFP 1G4 TCRα/β cells in the absence of CD8 coreceptor

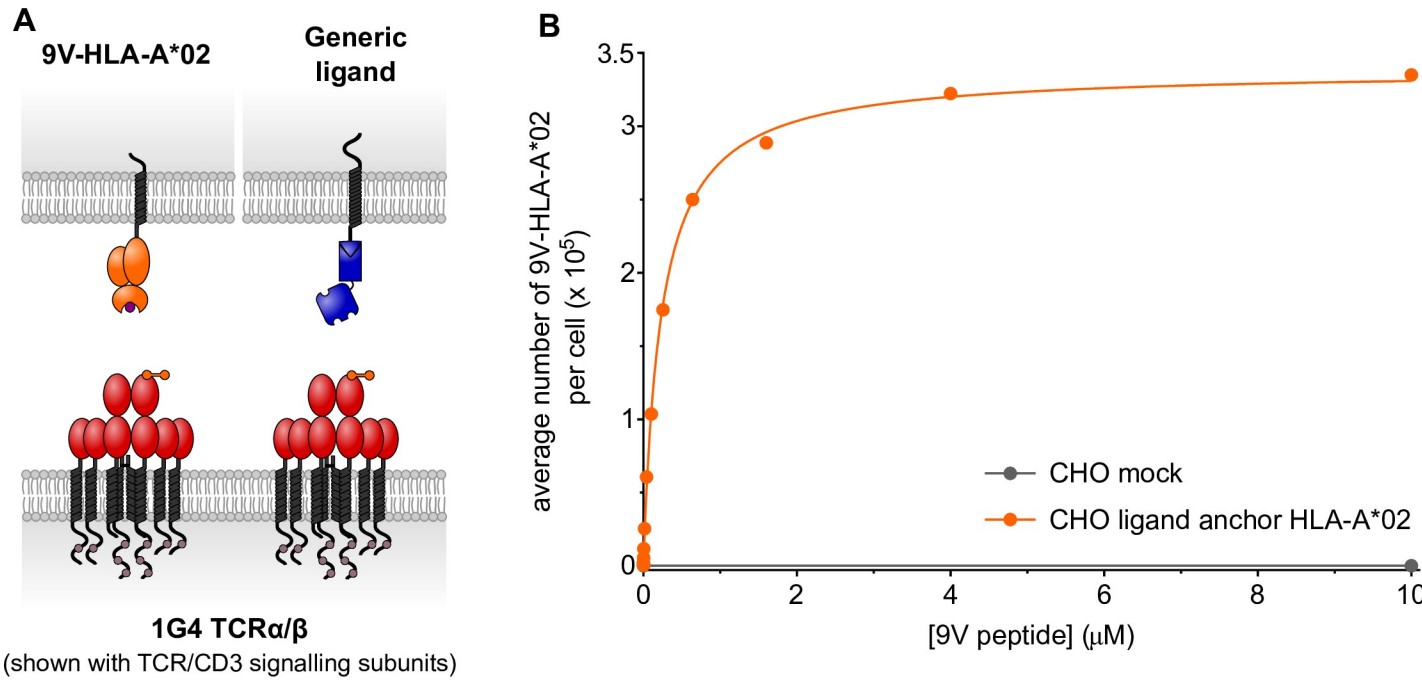

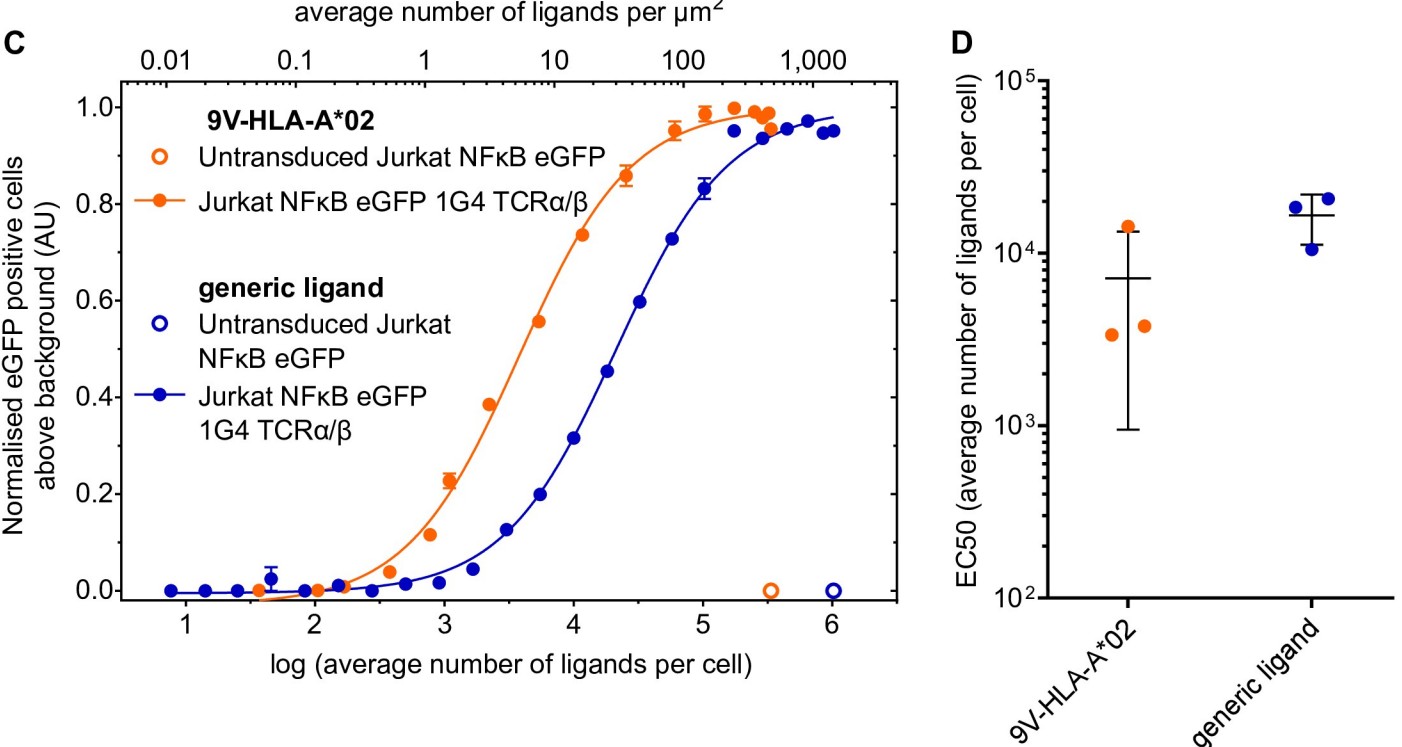

**Fig 4. 1G4 TCR responds to both generic and physiological ligand with a similar sensitivity.** (A) A cartoon depicting 1G4 TCR with β-chain N-terminal Twin-Strep-tag presented to either 9V-HLA-A*02 or generic ligand on CHO cells. (B) The average number of 9V-HLA-A*02 per cell as a function of peptide concentration added was measured using soluble fluorescent 1G4 high-affinity TCR and fluorescence quantitation beads. Interpolated numbers of 9V-HLA-A*02 are shown. (C) Response of Jurkat NFκB eGFP 1G4 TCRα/β cells to either generic ligand or 9V-HLA-A*02 presented on CHO cells. Receptor response, as indicated by eGFP expression under the control of the NFκB promoter, is shown normalised to the maximal receptor response to either 9V-HLA-A*02 or generic ligand. Error bars indicate the range ($n$ = 2), and data are representative of three independent experiments. (D) $EC_{50}$ values from individual experiments of Jurkat NFκB eGFP 1G4 TCRα/β cells responding to 9V-HLA-A*02 or generic ligand are plotted. Bars indicate the mean and standard deviation ($n$ = 3). Summary numerical data are provided in S1 Data. AU, arbitrary

units; CHO, Chinese hamster ovary; eGFP, enhanced green fluorescent protein; NFκB, nuclear factor kappa-light-chain-enhancer of activated B cells; TCR, T-cell receptor.

expression was very poor, we used cells that express CD8. Jurkat NFκB eGFP cells expressing either nontagged, Strep-tag II–tagged, or Twin-Strep-tagged 1G4 TCRα/β and CD8α/β were matched for TCRβ chain and CD8α expression (S4 Fig). These cells showed comparable cognate pMHC tetramer staining (S4 Fig), suggesting the presence of Strep-tag II does not significantly interfere with pMHC binding.

In summary, we have demonstrated that the generic ligand can elicit a receptor response comparable to physiological ligand, reinforcing its usefulness for studying receptor activation.

## Manipulating the generic ligand system

Our generic ligand system could be adapted to allow manipulation of ligand properties other than surface density, such as length, affinity, and valency, as illustrated in (Fig 5A–5C). For example, the receptor can be tagged with an N-terminal Strep-tag II and presented to monovalent Strep-Tactin-SpyCatcherΔ on cells, yielding a receptor–ligand pair with a 6-fold higher dissociation constant than the Twin-Strep-tag-trivalent Strep-Tactin pair, ($K_D$ = 43 μM) (Fig 5B, S5 Fig). Preparation of monovalent Strep-Tactin-SpyCatcherΔ and quantitation of monovalent Strep-Tactin-SpyCatcherΔ ligand numbers per cell are performed in the same manner as for the higher-affinity system (S5 Fig, S6 Fig). Siglec-14 with N-terminal Strep-tag II and expressed with its adaptor in THP-1 cells was able to respond to monovalent Strep-Tactin-SpyCatcherΔ presented on CHO cells (Fig 5D). The $EC_{50}$ of this response (mean value and standard deviation of 1,000,000 and 330,000 generic ligands per cell) to a lower-affinity interaction was higher, by approximately 5-fold, compared to that measured for Siglec-14 with Twin-Strep-tag and trivalent Strep-Tactin-SpyCatcher (mean $EC_{50}$ value and standard deviation of 190,000 and 170,000 generic ligands per cell) (Fig 3F, Fig 5E). Although this difference in $EC_{50}$ correlates with the difference in $K_D$, the expression levels of the Strep-tag II and Twin-Strep-tag receptors were not matched in these preliminary experiments.

In addition to monovalent Strep-Tactin, we presented the THP-1 Siglec-14 Strep-tag II cells to trivalent Strep-Tactin-SpyCatcher on CHO cells (Fig 5D). This increase in valency of the receptor–generic ligand interaction from monovalent to trivalent resulted in a 5-fold lower $EC_{50}$ of IL-8 secretion response by the THP-1 cells (mean $EC_{50}$ values of 200,000 versus 1,000,000 Strep-tag II–binding sites per cell for the trivalent and monovalent ligands, respectively). Although this shows that increased valency enhances activation of the Siglec-14 receptor, further experiments are required to determine to what extent this is the result of increasing receptor engagement or enhanced signalling.

The principles of the system can also be exploited to investigate poorly understood features of CARs such as requirements for their optimal signalling. There is growing interest in CARs with nanobody-based antigen-binding domains targeting tumour antigens (reviewed in [44]) [45, 46].

Here, we use standard first-generation CARs with either LaG17 or LaM8 nanobodies for ligand binding and TCRζ-chain intracellular region for signalling (LaG17-ζ and LaM8-ζ, Fig 6A and 6B). LaG17 and LaM8 bind to GFP and mCherry, respectively, with $K_D$ values of 50 nM and 63 nM (at 25°C) [13]. To form the CAR ligands, CHO ligand anchor cells were incubated with varying concentrations of soluble GFP or mCherry fused to SpyCatcherΔ (Fig 6A–6D). GFP or mCherry are thereby presented on CHO cell surfaces at a wide range of concentrations (Fig 6C and 6D) for engagement by the appropriate CAR.

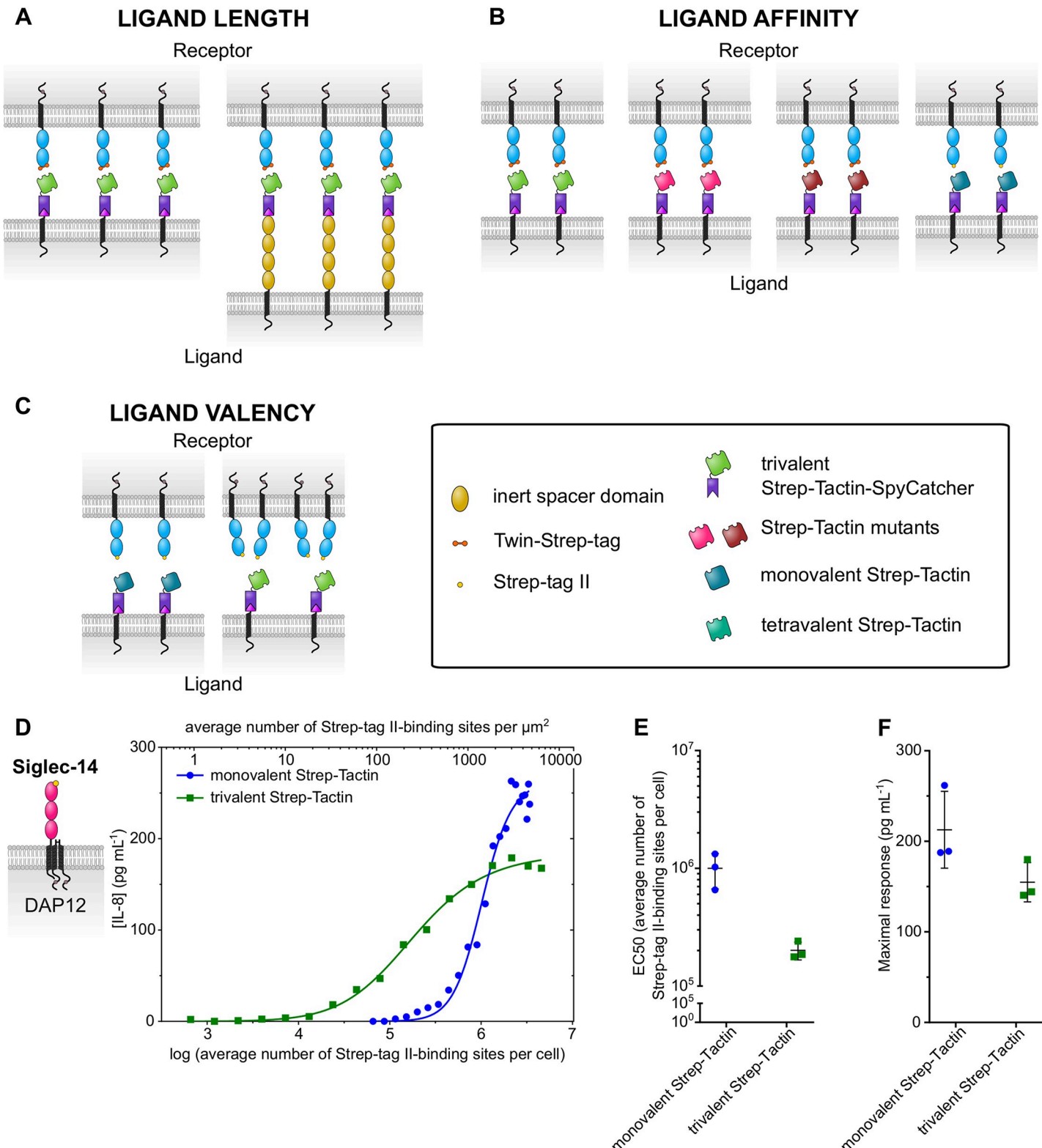

**Fig 5. Manipulations of the generic ligand system.** (A) The extracellular region of the generic ligand can be elongated by insertion of inert spacer domains. (B) Mutated forms of Strep-Tactin/streptavidin can be substituted into the receptor–generic ligand interaction to investigate how changing binding affinity affects receptor activation and signalling. Strep-Tactin variants are shown in different colours. (C) Strep-Tactin-SpyCatcher tetramers with varying numbers of Strep-tag II–binding sites can be coupled to CHO ligand anchor cells to examine the effect of varying the valency of the generic ligand on NTR triggering. For example, Strep-tag II–tagged

receptors can be presented to either monovalent or trivalent generic ligand. (D) Response of THP-1 Siglec-14–Strep-tag II DAP12 cells to monovalent Strep-Tactin or trivalent Strep-Tactin generic ligand presented on CHO cells. Receptor response is measured by IL-8 secretion. Data are representative of three independent experiments. Within each stimulation, a sample of CHO cells were taken and used to measure the number of Strep-tag II–binding sites per cell as in Fig 2. Ligand density was calculated from these numbers (see the Measuring generic ligand numbers per cell section). $EC_{50}$ (E) and maximal response values (F) from individual experiments of THP-1 Siglec-14–Strep-tag II DAP12 cells responding to monovalent Strep-Tactin or trivalent Strep-Tactin generic ligand are plotted. Bars indicate the mean and standard deviation ($n = 3$). Summary numerical data are provided in S1 Data. CHO, Chinese hamster ovary; DAP12, DNAX-activating protein of 12 kDa; IL-8, interleukin 8; NTR, non-catalytic tyrosine-phosphorylated receptor; Siglec-14, Sialic acid–binding immunoglobulin-type lectin 14.

Jurkat cells transduced to stably express either LaG17- ζ or LaM8-ζ were activated in a clear dose-dependent manner by CHO cells presenting GFP or mCherry, respectively (Fig 6E and 6F). This Jurkat cell response, indicated by up-regulation of CD69 surface expression, was specific. CD69 up-regulation in Jurkat LaG17-ζ cells was not seen in response to CHO mCherry ligand cells nor in Jurkat LaM8-ζ cells in response to CHO GFP ligand cells.

The efficiency of coupling and the high number of SpyTag ligand anchors suggested that the system could be extended to couple with multiple SpyCatcher fusion proteins to create cells presenting multiple ligands (Fig 6G). Indeed, we show by flow cytometry that CHO ligand anchor cells preincubated to present one level of GFP-SpyCatcherΔ can be subsequently incubated with varying concentrations of mCherry-SpyCatcherΔ to titrate the surface-presented levels of this second ligand (Fig 6H). These cells could be used, for example, in combination with cells presenting both LaG17-ζ and a receptor comprising the LaM8 nanobody fused to transmembrane and cytoplasmic regions of costimulatory or inhibitory receptors. This would facilitate analysis of signal integration between multiple receptors, which remains poorly understood.

## Discussion

We describe a novel generic ligand system that facilitates the investigation of receptors that bind cell surface ligands. This ligand is stably presented on the surface of CHO cells, the ligand density can be easily titrated over several orders of magnitude, and the surface density can be precisely controlled and measured.

Whereas other 'titratable' cell surface recognition systems have been described, none of these allow precise control and measurement of cell surface ligand density. For example, James and Vale used a system comprising FK506 binding protein (FKBP) and FKBP-binding domain (FRB), which only bind in the presence of rapamycin or analogue. The number of receptor–ligand interactions can be altered by the addition of different drug/analogue concentrations and permits fine temporal control over binding events [47, 48].

In addition, split, or 'switch-mediated', CAR systems have been described based on expression of a cell surface signalling component to which soluble antigen receptors or Fab fragments of diverse specificities can be coupled (reviewed in [49, 50]) [7]. Again, receptor–ligand interactions can be titrated by the addition of the soluble component.

Such systems are useful for flexible control over relative abundance of receptor–ligand interactions. However, to our knowledge, the surface density of ligands for recognition receptors has not been both varied and quantified in any of these systems. The combination of quantification and fine control of ligand surface density within a cellular environment offers a distinct advantage for studying receptor engagement and triggering. Such quantification, for example, permits the signalling properties revealed by studies using the generic ligand system to be considered within the context of the knowledge of native ligand densities.

We show that four representative NTRs can be stimulated by ligation of N-terminal Twin-Strep-tag. These results show that specific engagement of native ligand is not required for receptor triggering. This is in agreement with other work that shows these NTRs, among

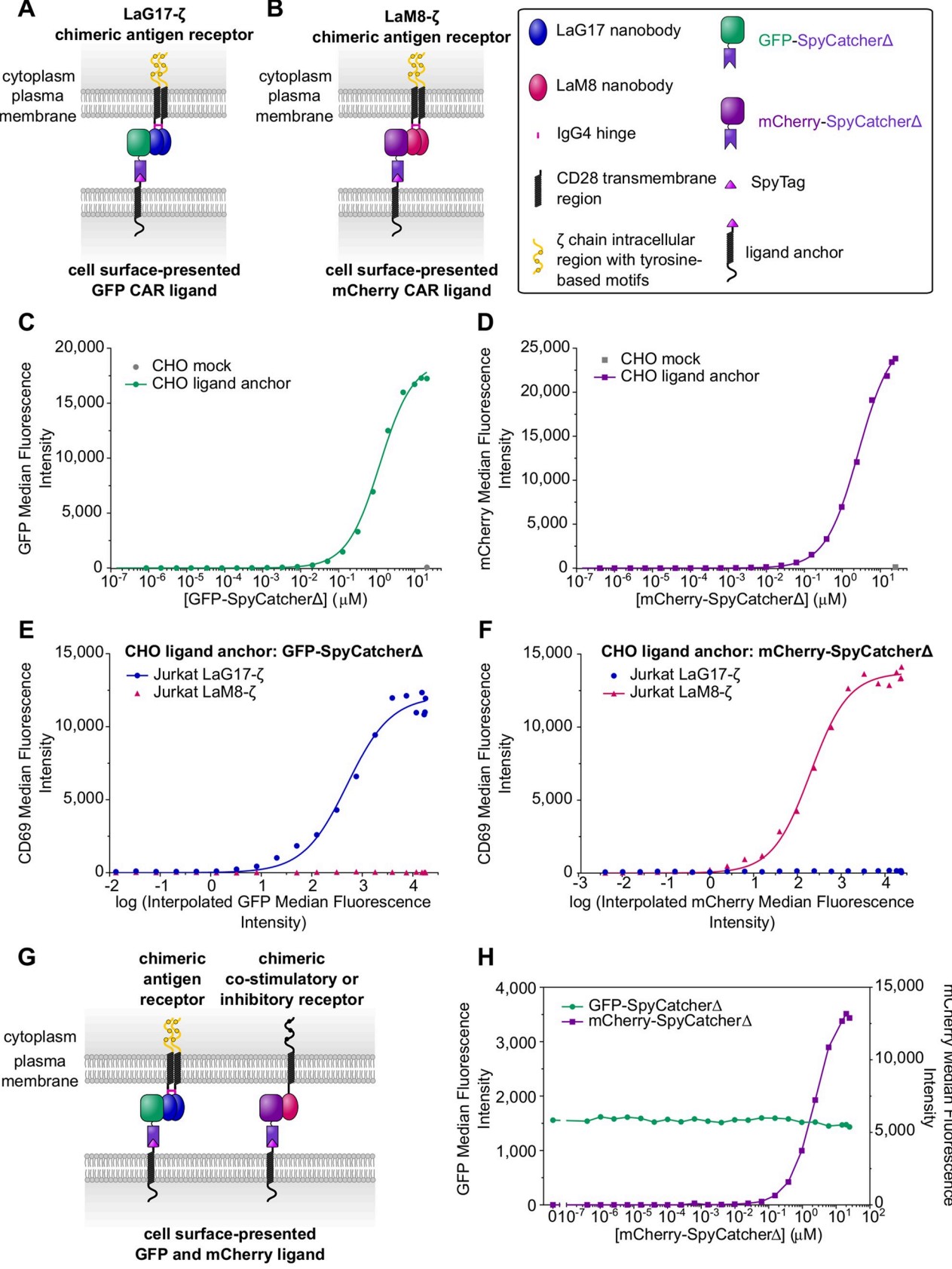

**Fig 6. Expanding the system to study CARs and signal integration.** (A-B) LaG17 anti-GFP and LaM8 anti-mCherry CARs are expressed in Jurkat cells. The CAR ligands comprise the cell surface–expressed ligand anchor with N-terminal SpyTag and soluble GFP-SpyCatcherΔ or mCherry-SpyCatcherΔ covalently coupled to the anchor. For simplicity, CARs are shown bound to a single ligand. (C-D) Amount of CAR ligand per CHO cell as a function of GFP-SpyCatcherΔ (C) or mCherry-SpyCatcherΔ (D) concentration added to cells. Median fluorescence intensity values extracted from flow cytometry analyses of cells are shown. (E-F) Response of Jurkat LaG17-ζ and Jurkat LaM8-ζ cells to either GFP-SpyCatcherΔ or mCherry-SpyCatcherΔ presented on CHO cells. Jurkat CD69 cell surface expression is plotted against the relative levels of the specific CAR ligand on CHO cells. These are the GFP or mCherry median fluorescence intensity values interpolated from the data shown in Fig 6C and 6D. Data are representative of two independent experiments. (G) To extend this system to study the requirements for optimal costimulatory and inhibitory receptor signalling, cells expressing the LaG17-ζ CAR and a fusion protein consisting of LaM8 nanobody followed by the transmembrane and intracellular regions of a costimulatory or inhibitory receptor can be presented to CHO ligand anchor cells presenting both GFP-SpyCatcherΔ and mCherry-SpyCatcherΔ at titratable levels. (H) CHO ligand anchor cells were first incubated with a single below-saturation concentration of GFP-SpyCatcherΔ and then titrating concentrations of mCherry-SpyCatcherΔ. Median fluorescence intensity values extracted from flow cytometry analyses of cells are shown. Summary numerical data are provided in S1 Data. CAR, chimeric antigen receptor; CHO, Chinese hamster ovary; GFP, green fluorescent protein; IgG4, immunoglobulin G4; mCherry, monomeric Cherry.

others, can be activated by antibodies to the receptor ectodomains coated on plastic (examples include [8, 37, 39]).

Twin-Strep-tagged SIRP $\beta$I, Siglec-14, and NKp30 respond to Strep-Tactin generic ligand with a similar sensitivity. The mean $EC_{50}$ values of the receptor responses from three independent experiments were 130,000–260,000 ligands per cell (Fig 3F). In contrast, the 1G4 TCR responds with a much lower mean $EC_{50}$ of 17,000 ligands per cell, indicating that TCR triggering required much lower surface densities of generic ligand than other NTRs (Fig 3F). Furthermore, the ligand densities of our generic ligand required to elicit responses were comparable to the required ligand densities of native ligands reported by others [47, 51–54].

A number of manipulations can be made to the generic ligand system to investigate the basic requirements and optimal conditions for NTR activation, as outlined in Fig 5A–5C. Increasing the extracellular dimensions of the generic ligand by inserting inert spacer domains into the anchor would enable testing whether ligand length affects recognition, as predicted by the kinetic-segregation model of NTR triggering (Fig 5A) [1, 55]. Until now, these experiments have been painstaking, as they require expressing different length forms of each native ligand at matched levels, and titration of ligand density is not possible [1, 56].

We also illustrate how the receptor–generic ligand affinity can be altered through using alternative SpyCatcher fusion proteins and/or variants of the Strep-tag II/Twin-Strep-tag system (Fig 5B and 5D, S5 Fig and S6 Fig). Thus, the sensitivity of receptors to affinity changes, and whether there is an optimal ligand binding affinity for various NTRs, can be explored in a high-throughput manner.

The $K_D$ values we measured for Twin-Strep-tag–mTFP binding to trivalent Strep-Tactin-SpyCatcher, and Strep-tag II–mTFP binding to monovalent Strep-Tactin-SpyCatcherΔ differ from previously reported measurements of either Strep-tag II or Twin-Strep-tag binding to tetravalent Strep-Tactin (Fig 2A, S5 Fig) [26, 28]. The relatively high $K_D$ value we measured for Strep-tag II binding to monovalent Strep-Tactin-SpyCatcherΔ is, however, in line with our observation that activation of Siglec-14 with N-terminal Strep-tag II requires very high surface densities of monovalent Strep-Tactin generic ligand (mean $EC_{50}$ of 1,000,000 generic ligands per cell, Fig 5E).

We also show how the role of receptor clustering in receptor triggering can be explored by varying the valency of the generic ligand (Fig 5C and 5D). This is achieved by refolding Strep-tag II binding and nonbinding (dead) Strep-Tactin subunits in varying ratios. Large-scale receptor clustering could also be induced using larger multivalent ligand complexes [10].

This variety of major receptor–ligand interaction properties that can be systematically altered, both alone and in combination, using the generic ligand system would not be as easily achievable, if at all, with the other flexible systems already described [7, 47–50]. Although

elements of modular CAR systems, such as described by Cho et al, could in principle be adapted to modulate ligand densities, such a system would have two disadvantages compared with the system described here. The valency of the ligand could not easily be varied, and its noncovalent capture would complicate precise control of its surface density [7].

Using GFP-SpyCatcherΔ and mCherry-SpyCatcherΔ, we were also able to apply this system to activate a first-generation CAR with an anti-GFP or anti-mCherry nanobody as the antigen-binding domain. Our generic ligand system should facilitate development of CARs and other synthetic receptors by allowing the direct, quantitative comparison of multiple receptors with varying design properties in a systematic and finely controlled manner.

The high density of ligand that can be achieved provides an opportunity to use more than one soluble SpyCatcher fusion protein simultaneously to yield combinations of multiple ligands that can be independently titrated (Fig 6H). For example, a second ligand could engage costimulatory or inhibitory NTRs or other immune receptors (Fig 6G). Such an approach greatly extends our ability to explore not just receptor activation but also signal integration in a powerful, multidimensional analysis. We note other methods could be employed to present multiple cell surface ligands to distinct receptors such as the leucine zipper pairs used by Cho and colleagues to study CAR signalling [7].

Alternative SpyCatcher fusion proteins could also be employed to study receptor signalling whilst maintaining exact receptor and native ligand binding properties. Specifically, a soluble fusion of a receptor's native ligand and SpyCatcher could be presented on generic ligand anchor cells. This would permit a more physiological approach but still within a cellular regime that is easily titratable and quantifiable.

The generic ligand system, and future work building upon foundations laid here, will aid both fundamental and translational research investigations into the processes that regulate immune signalling and signal integration. In principle, it can also be adapted to analyse the vast array of other receptor–cell surface ligand interactions that exist in a systematic, high-throughput manner.

## Supporting information

**S1 Fig. Preparation of trivalent Strep-Tactin-SpyCatcher.** (A) Trivalent Strep-Tactin-Spy-Catcher is synthesised by refolding mixtures of bacterially produced Strep-Tactin and dead streptavidin-SpyCatcher monomers in a 3:1 ratio. This desired tetramer is shown in schematic form alongside a more simplified cartoon. (B) An anion exchange chromatogram showing elution of the predicted trivalent Strep-Tactin-SpyCatcher peak alongside other configurations including a monovalent Strep-Tactin tetramer with a single Strep-tag II–binding site and three SpyCatchers. (C) SDS-PAGE analysis of the eluted anion exchange chromatography peaks predicted to contain trivalent Strep-Tactin-SpyCatcher and monovalent Strep-Tactin. The samples were boiled prior to loading, and the gel was stained with Coomassie to show the relative proportion of subunits. Densitometry was performed on Strep-Tactin and dead streptavidin-SpyCatcher bands, and the values were normalised for subunit molecular weights and then converted into a Strep-Tactin:dead streptavidin-SpyCatcher subunit ratio. Trivalent Strep-Tactin-SpyCatcher = 4.7:1 (expected 3:1), monovalent Strep-Tactin peak = 0.6:1 (expected 0.33:1). (D) Trivalent Strep-Tactin-SpyCatcher or the predicted monovalent peak for comparison (50 nM) was incubated with a titration of biotin-4-fluorescein in a fluorescence-quenching assay. Inflection point X values (X0) are shown. Summary numerical data are provided in S1 Data; original gel images are provided in S1 Raw images.
(TIF)

**S2 Fig. Optimal conditions for generating the complete generic ligand.** (A) Ligand anchor expression by transfected CHO cells as determined using antibody to N-terminal HA tag and flow cytometry (CHO mock: mock transfected). The efficiency of CHO ligand anchor:trivalent Strep-Tactin-SpyCatcher coupling at 25˚C under different pH conditions (B) or with varying cell–protein incubation times before washing (C) is shown. Cells were incubated with ATTO 647 biotin to indicate generic ligand levels. MFI values, extracted from flow cytometry analyses, are shown as a function of trivalent Strep-Tactin-SpyCatcher concentration. (D) Ligand anchor:trivalent Strep-Tactin-SpyCatcher binding is covalent. Boiled lysates of CHO ligand anchor cells preincubated with trivalent Strep-Tactin-SpyCatcher or buffer only were analysed by western blotting. The Strep-Tactin-SpyCatcher tetramer dissociates upon boiling, and so the ligand anchor is visualised coupled to dead streptavidin-SpyCatcher subunit only. (E) Cell surface down-regulation of the generic ligand over time following reconstitution is visualised using ATTO 647 biotin. MFIs, extracted from flow cytometry analyses, are shown normalised to the MFI at time 0, which was given a value of 1. The mean half-life from two independent experiments (range = 780–860 minutes, $n = 2$) is shown. The generic ligand cell surface levels appear to rise within the first 20 minutes post-reconstitution, visualised as an increase in MFI. This may reflect a proportion of trivalent Strep-Tactin-SpyCatcher that is in contact with, but not yet covalently bound to, ligand anchor during the initial incubation and so is removed during the process of analysing ligand cell surface levels. Incubating the cells at 37˚C post-reconstitution may allow this proportion of protein to covalently, irreversibly bind to the ligand anchor and thereby lead to an apparent increase in cell surface levels. Summary numerical data are provided in S1 Data; gating strategy and original .fcs files are provided in S2 Data; original gel images are provided in S1 Raw images. CHO, Chinese hamster ovary; HA, hemagglutinin; IB, immunoblot; MFI, median fluorescence intensity. (TIF)

**S3 Fig. Twin-Strep-tagged receptor and associated adaptor expression and CHO ligand anchor HLA-A*02 expression.** (A) Expression of one of three Twin-Strep-tagged receptors and appropriate adaptor by THP-1 cells. Using flow cytometry, receptor expression was analysed using anti-Strep-tag II antibody. Expression of the exogenous, introduced adaptor was inferred using an IRES-EmGFP sequence. (B) Expression of 1G4 TCRα/β-Twin-Strep-tag by Jurkat NFκB eGFP cells as shown using anti-Strep-tag II antibody and flow cytometry. (C) Expression of the generic ligand anchor and HLA-A*02 SCD by CHO cells shown using anti-HA tag antibody and anti-HLA-A*02 antibody, respectively. Numbers indicate percentage of events in each quadrant. Gating strategy and original .fcs files in S2 Data. CHO, Chinese hamster ovary; eGFP, enhanced green fluorescent protein; HA, hemagglutinin; IRES-EmGFP, internal ribosome entry site–emerald green fluorescent protein; NFκB, nuclear factor kappa-light-chain-enhancer of activated B cells; SCD, single-chain dimer; TCR, T-cell receptor. (TIF)

**S4 Fig. Quantification of 9V-HLA-A*02 per cell and demonstration that Twin-Strep-tag does not interfere with TCR-9V-HLA-A*02 binding.** (A) Median fluorescence intensity values from flow cytometry analysis of Alexa Fluor 647 fluorescence quantitation beads used to create a standard curve. (B) A relative indication of the level of 9V-HLA-A*02 per cell as a function of 9V peptide concentration added to cells. Median fluorescence intensity values extracted from flow cytometry analyses of cells incubated with soluble 1G4 high-affinity TCRα/β Alexa Fluor 647 are shown. (C) Jurkat reporter cells expressing CD8α and β and 1G4 TCRα/β either nontagged or tagged with Strep-tag II or Twin-Strep-tag show comparable levels of TCRβ chain (left) and CD8α (centre) expression and 9V-HLA-A*02 tetramer binding (right). Summary numerical data are provided in S1 Data; gating strategy and original .fcs files

are in S2 Data. TCR, T-cell receptor.
(TIF)

**S5 Fig. Lowering receptor–generic ligand affinity using Strep-tag II–tagged receptor and monovalent Strep-Tactin-SpyCatcherΔ.** (A) (1) The receptor is constructed with N-terminal Strep-tag II. (2) Soluble monovalent Strep-Tactin-SpyCatcherΔ protein covalently binds to the generic ligand anchor. (3) The single binding site of monovalent Strep-Tactin-SpyCatcherΔ is available for ligation by the Strep-tag II–tagged receptor. (B) Monovalent Strep-Tactin-Spy-CatcherΔ is synthesised by refolding mixtures of bacterially produced Strep-Tactin-Spy-CatcherΔ and dead streptavidin monomers in a 1:3 ratio. (C) An anion exchange chromatogram showing elution of the predicted monovalent Strep-Tactin-SpyCatcherΔ peak alongside other configurations. (D) SDS-PAGE analysis of the eluted anion exchange chromatography peak predicted to contain monovalent Strep-Tactin-SpyCatcherΔ. Densitometry was performed on the bands, and the values were normalised for subunit molecular weights and then converted into a Strep-Tactin-SpyCatcherΔ:dead streptavidin subunit ratio (1:3.7, expected 1:3). (E) Monovalent Strep-Tactin-SpyCatcherΔ (50 nM) was incubated with a titration of biotin-4-fluorescein in a fluorescence-quenching assay. The inflection point (X0) is shown. (F) Representative equilibrium binding from surface plasmon resonance of Strep-tag II–mTFP flown over immobilised monovalent Strep-Tactin-SpyCatcherΔ at 37˚C. The $K_D$ (SEM) for the collated data from three independent experiments with two flow cells per experiment is 43 μM (4.5 μM), and the mean Hill slope (SEM) is 0.97 (0.085) to 2 s.f. Summary numerical data are provided in S1 Data; original gel images are provided in S1 Raw images. mTFP, monomeric teal fluorescent protein; SEM, standard error of the mean; s.f., significant figures.
(TIF)

**S6 Fig. Quantification of monovalent Strep-Tactin-SpyCatcherΔ proteins per CHO ligand cell.** (A) A relative indication of the level of generic ligand per cell as a function of monovalent Strep-Tactin-SpyCatcherΔ concentration. Geometric mean fluorescence intensity values from flow cytometry analyses of cells incubated with ATTO 488 biotin are shown. (B) CHO ligand anchor cells preincubated with monovalent Strep-Tactin-SpyCatcherΔ or buffer alone were incubated with a titration of biotin-4-fluorescein in a fluorescence-quenching assay. (C) Expression of Siglec-14-Strep-tag II and exogenous DAP12 by THP-1 cells using anti-Strep-tag II antibody and an IRES-EmGFP sequence respectively and flow cytometry. Percentages of events in each quadrant are shown. Summary numerical data are provided in S1 Data; gating strategy and original .fcs files in S2 Data. CHO, Chinese hamster ovary; DAP12, DNAX-activating protein of 12 kDa; IRES-EmGFP, internal ribosome entry site–emerald green fluorescent protein; Siglec-14, Sialic acid–binding immunoglobulin-type lectin 14.
(TIF)

**S7 Fig. Quantification of generic ligands per CHO ligand cell using both biotin-4-fluorescein and quantitation beads.** (A) CHO ligand anchor cells presenting either trivalent streptavidin-SpyCatcher or trivalent Strep-Tactin-SpyCatcher were analysed for anti-streptavidin antibody binding. Median fluorescence intensities from flow cytometry analyses are shown. (B) CHO ligand anchor cells preincubated with trivalent Strep-Tactin-SpyCatcher (left) or trivalent streptavidin-SpyCatcher (right) were incubated with a titration of biotin-4-fluorescein in a fluorescence-quenching assay. The average numbers of generic ligands per cell calculated using the X0 values are shown. (C) Anti-mouse IgG quantitation beads were incubated with anti-streptavidin antibody. Median fluorescence intensities from flow cytometry analyses are shown plotted against the antibody binding capacity of the beads. (D) CHO ligand anchor

cells preincubated with trivalent streptavidin-SpyCatcher or buffer alone were incubated with anti-streptavidin antibody in parallel with the anti-mouse IgG beads in (C) and analysed by flow cytometry. The median fluorescence intensity value is shown alongside the number of ligands per cell calculated using this value and the standard curve in (C). Ligand numbers are given to 2 s.f. Summary numerical data are provided in S1 Data; gating strategy and original .fcs files are in S2 Data. CHO, Chinese hamster ovary; IgG, immunoglobulin G; s.f., significant figures.
(TIF)

**S1 Data. Contains data pertaining to** Fig 2A, Fig 2B, Fig 2C, Fig 2D, Fig 3B, Fig 3C, Fig 3D, Fig 3E, Fig 3F, Fig 4B, Fig 4C, Fig 4D, Fig 5D, Fig 5E, Fig 5F, Fig 6C, Fig 6C, Fig 6D, Fig 6E, Fig 6F, Fig 6H, S1 Fig, S2 Fig, S4 Fig, S5 Fig, S6 Fig, S7 Fig.
(XLSX)

**S2 Data. Contains flow cytometry gating strategy and original .fcs files for flow cytometry data shown in** S2 Fig, S3 Fig, S4 Fig, S6 Fig, S7 Fig.
(ZIP)

**S1 Raw images. Contains original gel and blot images for data shown in** S1 Fig, S2 Fig, S5 Fig. Vertical white lines indicate where images were spliced to remove irrelevant lanes. Image colours were inverted and brightness of entire image as a whole was altered prior to creating figures.
(PDF)

## Acknowledgments

We acknowledge Mark Howarth for providing Strep-Tactin, streptavidin, dead streptavidin, SpyTag, and SpyCatcher constructs and for helpful discussions and advice; Peter Steinberger and Wolfgang Paster for providing Jurkat reporter cell lines; Oreste Acuto for providing 1G4 TCR construct; and Adaptimmune Limited for providing the c58c61 TCR. We also acknowledge William J. Barton for contributions to the experimental data shown in Fig 5D–5F. We thank Mikhail Kutuzov for providing 9V-HLA-A*02 and other members of the Dushek group for providing 9V-HLA-A*02 tetramers. We also thank Marion H. Brown and Omer Dushek for helpful discussion.

## Author Contributions

**Conceptualization:** P. Anton van der Merwe, Jesse Goyette.

**Data curation:** Eleanor M. Denham, Jesse Goyette.

**Formal analysis:** Eleanor M. Denham, P. Anton van der Merwe, Jesse Goyette.

**Funding acquisition:** P. Anton van der Merwe, Jesse Goyette.

**Investigation:** Eleanor M. Denham, Michael I. Barton, Susannah M. Black, Marcus J. Bridge, Ben de Wet, Rachel L. Paterson, Jesse Goyette.

**Methodology:** Rachel L. Paterson, Jesse Goyette.

**Supervision:** P. Anton van der Merwe, Jesse Goyette.

**Writing – original draft:** Eleanor M. Denham.

**Writing – review & editing:** Eleanor M. Denham, P. Anton van der Merwe, Jesse Goyette.

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
