## [Editor Report · Decision Letter 0]

5 Sep 2019

Dear Dr Goyette, 

Thank you for submitting your manuscript entitled "A generic cell surface ligand system for studying cell-cell recognition" for consideration as a Methods and Resources by PLOS Biology.

I have assessed your revised manuscript and consulted with my colleagues. I am writing to let you know that we would like to send your submission back out for external peer review.

*Please be aware that, due to the voluntary nature of our reviewers and academic editors, manuscripts may be subject to delays during the holiday season. Thank you for your patience.*

Please re-submit your manuscript within two working days, i.e. by Sep 07 2019 11:59PM.

Kind regards,

Lauren A Richardson, Ph.D

Senior Editor

PLOS Biology

---

## [Decision Letter · Decision Letter 1]

16 Oct 2019

Dear Dr Goyette,

Thank you for submitting your revised Methods and Resources entitled "A generic cell surface ligand system for studying cell-cell recognition" for publication in PLOS Biology. I have now obtained advice from the original reviewers and have discussed their comments with the Academic Editor. 

Based on the reviews, we will probably accept this manuscript for publication, assuming that you will modify the manuscript to address the remaining concerns raised by the reviewers. Specifically, we ask that you cite and discuss the articles mentioned by Rev #3.

We expect to receive your revised manuscript within two weeks. Your revisions should address the specific points made by each reviewer. In addition to the remaining revisions and before we will be able to formally accept your manuscript and consider it "in press", we also need to ensure that your article conforms to our guidelines. A member of our team will be in touch shortly with a set of requests. As we can't proceed until these requirements are met, your swift response will help prevent delays to publication.

Please note that you may have the opportunity to make the peer review history publicly available. The record will include editor decision letters (with reviews) and your responses to reviewer comments. If eligible, we will contact you to opt in or out.

Sincerely,

Lauren A Richardson, Ph.D

Senior Editor

PLOS Biology

DATA POLICY:

Regardless of the method selected, please ensure that you provide the individual numerical values that underlie the summary data displayed in the following figure panels: (e.g. Figs. 2A-D, 3B-F, 4B-D, 5D-F, 6CDEFH, S1D, S2BCE, S4AB, S5EF, S6AB), as they are essential for readers to assess your analysis and to reproduce it. 

For figures containing FACS data (S2A, S3, S4C, S6C), we ask that you provide FCS files and a picture showing the successive plots and gates that were applied to the FCS files to generate the figure.

Please also ensure that figure legends in your manuscript and your Data Statement in the submission system accurately describe where your data can be found.

For manuscripts submitted on or after 1st July 2019, we require the original, uncropped and minimally adjusted images supporting all blot and gel results reported in an article's figures or Supporting Information files. We will require these files before a manuscript can be accepted so please prepare them now, if you have not already uploaded them. Please carefully read our guidelines for how to prepare and upload this data: https://journals.plos.org/plosbiology/s/figures#loc-blot-and-gel-reporting-requirements.

Reviews [reviewer numbering maintained from first round]

Reviewer #1: 

Reviewer's comments appear to have been addressed appropriately.

--------------

Reviewer #2: 

The authors were able to address my major concerns.

1) They have convinced me that a generic ligand can be informative even it it does not mimic aspects of the natural receptor:ligand geometry. This approach as with other systems like it can decouple the potential signaling capability of the receptor from the natural binding constraints. This is useful in synthetic biology where there is a desire to co-opt and engineer natural systems for therapeutic purposes.

They also point out that having a generic ligand system can provide a clue to whether the natural binding geometry and dynamics are a major contributor to the overall signaling and cellular outcome. For example, in the hypothetical case where their generic ligand system exhibits major differences from what has been observed for the natural system. This would suggest the natural binding features are critical to the biology. However, they haven't provided a case of this here.

2) They have showcased the versatility of their approach by providing data on the effects of ligand valency in figure 5.

3) I find their chimeric antigen receptor work interesting, but the data is minimal. This is a methods paper so I am okay with this. Overall, the approach could be interesting to use to study different types of signaling domains individually or in concert. 

In the end, this approach may be most useful for synthetic biology purposes where we seek to simultaneously understand how we can gain control over activation properties of a receptor system and at the same time be able to retarget the receptor to any ligand of interest. I would suggest adding a passage on this to manuscript.

--------------

Reviewer #3: 

My biggest concern with this work is its differentiation over existing work, especially with regard to the split CAR systems that have been developed recently. The author listed 3 things that set themselves apart from existing work. However, most of the features that the authors highlighted as unique have been achieved elsewhere.

1. The authors suggest that their system allows variation in receptor ligand affinity, valancey and size. However, this can be all be achieved by split CAR design by changing the antibody affinity, number of antibody domain, or changing the size of the binding domain. Please see Cho et al, Cell 2018 (ref 7 in this manuscript) for an example of ligand affinity modulation. While these published articles didn't compare the size of the adapter molecules, I cannot see why this is not achievable with other systems. 

2. The authors suggest that their system affords accurate quantification of the number of ligands, implying that there are not other methods. NIST has been developing standards using antibody and flow cytometry to quantify surface molecules. This is now the standard approach to obtain absolute measurement of surface molecules. It is hard for me to agree that the author's approach is better than what NIST has been developing, especially considering the serious restriction that this system impose over the use of fluorescent-labeled antibody.

Cytometry A. 2012 Jul;81(7):567-75. doi: 10.1002/cyto.a.22060. Epub 2012 Apr 26.

Human CD4+ lymphocytes for antigen quantification: characterization using conventional flow cytometry and mass cytometry.

Wang L1, Abbasi F, Ornatsky O, Cole KD, Misakian M, Gaigalas AK, He HJ, Marti GE, Tanner S, Stebbings R.

3. The authors suggests that their system affords multiple ligands to be easily added. Again, please see Cho et al, Cell, 2018 for a published example of combinatorial split CAR system that targets 2 ligands in human primary T cells.

Based on these facts, I am still not convinced that this work is sufficiently novel over existing work. The major difference between this work and existing split CAR design is the covalent linkages vs. antibody binding (which can have a very high affinity that approaches covalent linkages). The authors imply that covalent linkages afford unique properties that other systems cannot achieve, such as modulation in valency, ligand affinity, and quantification of ligand density, which is patently false.

--------------

Reviewer #4: 

The authors have satisfactorily addressed my concerns.

---

## [Editor Report · Decision Letter 2]

12 Nov 2019

***EDIT THIS LETTER BEFORE SENDING***

Dear Dr Goyette,

On behalf of my colleagues and the Academic Editor, Philippa Marrack, I am pleased to inform you that we will be delighted to publish your Methods and Resources in PLOS Biology. 

Early Version

PRESS 

Kind regards,

Sofia Vickers

Senior Publications Assistant

PLOS Biology

On behalf of, 

Lauren Richardson,

Senior Editor

PLOS Biology